# FNIP1 abrogation promotes functional revascularization of ischemic skeletal muscle by driving macrophage recruitment

Zongchao Sun [1,7], Likun Yang[1,7], Abdukahar Kiram[1,7], Jing Yang[1,7], Zhuangzhuang Yang[2], Liwei Xiao[1], Yujing Yin[1], Jing Liu[1], Yan Mao[1], Danxia Zhou[1], Hao Yu[1], Zheng Zhou[1], Dengqiu Xu[1], Yuhuan Jia[1], Chenyun Ding[1], Qiqi Guo[1], Hongwei Wang [3], Yan Li [4], Li Wang[4], Tingting Fu [1] ✉, Shijun Hu [2] ✉ & Zhenji Gan [1,5,6] ✉

Ischaemia of the heart and limbs attributable to compromised blood supply is a major cause of mortality and morbidity. The mechanisms of functional angiogenesis remain poorly understood, however. Here we show that FNIP1 plays a critical role in controlling skeletal muscle functional angiogenesis, a process pivotal for muscle revascularization during ischemia. Muscle FNIP1 expression is down-regulated by exercise. Genetic overexpression of FNIP1 in myofiber causes limited angiogenesis in mice, whereas its myofiber-specific ablation markedly promotes the formation of functional blood vessels. Interestingly, the increased muscle angiogenesis is independent of AMPK but due to enhanced macrophage recruitment in FNIP1-depleted muscles. Mechanistically, myofiber FNIP1 deficiency induces PGC-1α to activate chemokine gene transcription, thereby driving macrophage recruitment and muscle angiogenesis program. Furthermore, in a mouse hindlimb ischemia model of peripheral artery disease, the loss of myofiber FNIP1 significantly improved the recovery of blood flow. Thus, these results reveal a pivotal role of FNIP1 as a negative regulator of functional angiogenesis in muscle, offering insight into potential therapeutic strategies for ischemic diseases.

Skeletal muscle ischemia attributable to compromised blood supply is a common problem in cardiovascular and metabolic diseases and the leading cause of limb amputations[1–4]. A decline in skeletal muscle capillary density and blood flow also occurs during aging and is associated with the main cause of mortality and morbidity[5,6]. Exercise is an effective therapy for microvascular diseases and peripheral arterial diseases by promoting muscle functional angiogenesis—the formation of patent, nonleaky, and pericyte-covered blood vessels. However, mechanisms of functional angiogenesis in skeletal muscle remain poorly understood.

[1]State Key Laboratory of Pharmaceutical Biotechnology and MOE Key Laboratory of Model Animal for Disease Study, Model Animal Research Center, Division of Spine Surgery, Department of Orthopedic Surgery, Nanjing Drum Tower Hospital, Medical School of Nanjing University, Nanjing University, Nanjing, China. [2]Department of Cardiovascular Surgery of the First Affiliated Hospital & Institute for Cardiovascular Science, Collaborative Innovation Center of Hematology, State Key Laboratory of Radiation Medicine and Protection, Suzhou Medical College, Soochow University, Suzhou, China. [3]Center for Translational Medicine and Jiangsu Key Laboratory of Molecular Medicine, Medical School of Nanjing University, Nanjing, China. [4]State Key Laboratory of Food Science and Technology, School of Food Science and Technology, Jiangnan University, Wuxi, China. [5]Jiangsu Key Laboratory of Molecular Medicine, Medical School of Nanjing University, Nanjing University, Nanjing, China. [6]Chemistry and Biomedicine Innovation Center (ChemBIC), Nanjing University, Nanjing, China. [7]These authors contributed equally: Zongchao Sun, Likun Yang, Abdukahar Kiram, Jing Yang. ✉e-mail: futt@nicemice.cn; shijunhu@suda.edu.cn; ganzj@nju.edu.cn

Skeletal muscle functional revascularization from ischemia is a highly coordinated complex process involving interactions between several cell types and signals within the muscle microenvironment, including myofibers, endothelial cells, and macrophages[7–9]. A number of myofiber proangiogenic factors are known to coordinate endothelial cell activation, proliferation, migration and pericyte recruitment in skeletal muscle[10,11]. Macrophages are another essential components of functional angiogenesis in skeletal muscle, and this is particularly true after ischemia[7,8,12]. Macrophages undergo pro-inflammatory M1- to M2-like phenotype conversion to promote skeletal muscle angiogenesis remodeling[7,10,13]. Previous studies have revealed that multiple transcriptional factors such as nuclear receptor ERRs and HIF, along with coregulators PGC-1s control the expression of myofiber angiogenic growth factors and regulate muscle angiogenesis[14–18]. Another proangiogenic regulator with a strong impact on skeletal muscle angiogenesis is the metabolic regulator AMP-activated protein kinase (AMPK)[16,19]. A better understanding of the complex interplays between the intrinsic myofiber signaling network and microenvironmental cues will uncover the molecular regulatory mechanisms of functional angiogenesis in skeletal muscle, thus facilitating the development of effective medical therapy for ischemic diseases.

Folliculin interacting protein 1 (FNIP1) is an adaptor protein originally identified through its interaction with folliculin (FLCN) and AMPK[20]. While the FNIP1 protein has been implicated in regulating cellular metabolism through influencing metabolic sensors AMPK and mTOR signaling in multiple cell types[20–26], emerging evidence underpins that FNIP1 can exert regulatory roles in directing context-dependent cellular processes, including reductive stress, protein stability and calcium dynamics, beyond AMPK or mTOR signaling[27–31]. For instance, recent studies have demonstrated that FNIP1 could regulate the core machinery of the reductive stress response, chaperone function of Hsp90 and the activity of ER $Ca^{2+}$-ATPase SERCA[28,29,31]. A growing body of evidence has revealed implications of FNIP1 protein in the control of B-cell development, cardiomyopathy, kidney development and function, mitochondrial function muscle fiber type, and adipocyte browning[20,22–24,26,31–35]. Further underscoring the pathological importance of the FNIP1, mutations in FNIP1 are recognized to cause severe B-cell development defect, agammaglobulinemia, hypertrophic cardiomyopathy and pre-excitation syndrome in humans[23,36]. However, it has yet to be unraveled whether FNIP1 signaling is implicated in controlling functional angiogenesis in skeletal muscle, and especially in the setting of ischemia.

In this work, we explored the potential of FNIP1 in regulating the angiogenesis remodeling in skeletal muscle. We found that muscle FNIP1 expression is downregulated in response to exercise training. Using both gain-of-function and loss-of-function cell type-specific genetic models, we demonstrated that myofiber FNIP1 is a powerful negative regulator of functional angiogenesis in skeletal muscle. Myofiber-specific ablation of FNIP1 markedly enhanced the formation of patent, functional blood vessels. Mechanistically, we demonstrated that the increased muscle angiogenesis is independent of AMPK but due to enhanced macrophage recruitment in FNIP1-depleted muscles. Myofiber FNIP1 deficiency induces PGC-1α to activate chemokine gene transcription, thereby driving the macrophage recruitment and muscle angiogenesis program. Importantly, the deletion of myofiber FNIP1 improved the recovery of blood flow in the murine hindlimb ischemia model of peripheral artery disease. Together, our results demonstrate a previously unappreciated FNIP1-macrophage regulatory mechanism in the control of functional angiogenesis in skeletal muscle.

## Results

### FNIP1-dependent regulation of skeletal muscle angiogenesis
Exercise training is effective in stimulating angiogenesis in skeletal muscle. We showed that both the mRNA and protein expression of *Fnip1* were reduced in skeletal muscle of the swimming exercise training mice (Fig. 1a). We also conducted in situ hybridization to stain *Fnip1* mRNA within skeletal muscles and found that *Fnip1* mRNA is expressed in both myofibers and capillaries (Supplementary Fig. 1a). After an acute bout of exercise, expression of the *Fnip1* mRNA in situ hybridization was also reduced in both myofibers and capillaries (Supplementary Fig. 1a). We have previously generated transgenic animals that express *Fnip1* specifically in myofiber using the muscle creatine kinase promoter (MCK-FNIP1 Tg)[26]. The MCK-*Fnip1* transgene transcript was efficiently expressed in skeletal muscle, and we observed no change in *Fnip1* mRNA levels in endothelial cells (EC) isolated from MCK-FNIP1 Tg mice compared to non-transgenic (NTG) controls (Supplementary Fig. 1b). To test whether FNIP1 can regulate skeletal muscle angiogenesis, transverse frozen sections of gastrocnemius (GC) muscles from MCK-FNIP1 Tg and NTG mice were firstly evaluated by immunofluorescent staining for CD31, an endothelial marker that is commonly used to examine tissue angiogenesis and capillary density. We found that MCK-FNIP1 Tg muscles showed decreased CD31 staining compared to littermate NTG controls (Fig. 1b, c). We also evaluated whether overexpression of FNIP1 in myofiber exerts a regulatory action in EC proliferation or cell death. Ki67 staining showed that there was no significant difference in the number of Ki67$^+$CD31$^+$ cells in skeletal muscles between MCK-FNIP1 Tg and NTG mice (Supplementary Fig. 1c). However, TUNEL assays showed that overexpression of FNIP1 in myofiber leads to increases EC cell apoptosis in skeletal muscles from MCK-FNIP1 Tg mice (Supplementary Fig. 1d). Together, these findings point toward a possible suppression of skeletal muscle angiogenesis by myofiber FNIP1.

Using the FNIP1-knockout (FNIP1 KO) mice and the mice expressing FNIP1 only in skeletal muscle (FNIP1 TgKO mice) generated by crossing the MCK-FNIP1 Tg mice with FNIP1 KO mice, in situ hybridization confirmed the efficient deletion of *Fnip1* in skeletal muscle by KO, and the MCK-*Fnip1* transgene transcript was expressed in a myofiber-specific manner (Supplementary Fig. 1e). We have recently uncovered the role of FNIP1 in controlling skeletal muscle color and mitochondrial function muscle fiber type[26] (Fig. 1d). We speculated that loss of FNIP1 could simultaneously regulate muscle vessel formation to match the increased oxygen and nutrient delivery demand of type I muscle fibers. To test this, we further analyzed the gene expression data generated from *Fnip1*$^{+/+}$, FNIP1 KO, and FNIP1 TgKO muscles (Supplementary Fig. 2a). Notably, gene ontology (GO) analysis revealed that the primary FNIP1-regulated genes belong to angiogenesis process (Fig. 1e). A broad array of genes involved in angiogenesis program was induced in FNIP1 KO muscles (Supplementary Fig. 2b). We confirmed that key biomarker genes associated with muscle angiogenesis (*Vegfa*, *Vegfa120*, *Vegfa164*, *Vegfa188*, *Vegfb167*, *Vegfb186*, *Esm1*, and *Nrp2*) were significantly upregulated in the GC muscles of the FNIP1 KO mice but reduced in FNIP1 TgKO mice (Fig. 1f). Moreover, the density of capillaries was induced markedly in GC muscles of the FNIP1 KO mice but reduced in FNIP1 TgKO mice, as determined by CD31 staining (Fig. 1g, h). Hence, loss of muscle FNIP1 potently induces angiogenesis in skeletal muscle in vivo.

We next sought to examine whether FNIP1 deficiency induces the formation of functional blood vessels in skeletal muscle, we first used microangiography following intraventricular perfusion of fluorescent microspheres (0.1 μm) to visualize perfused microvasculature. The impermeability of the microspheres allows their vascular retention, enabling imaging of intact and mature muscle blood vessels. Examination of perfused microspheres in GC muscles revealed a marked increase in the number of patent vessels in FNIP1 KO mice but reduced in FNIP1 TgKO mice (Fig. 1g, h and Supplementary Fig. 2c, d), indicating that the blood vessels induced by FNIP1 deficiency are patent and capable of sustaining blood flow. Similar observations were made in white vastus lateralis (WV) muscles (Supplementary Fig. 2c, d). In addition, we also conducted an Evans blue (EB) dye infiltration test, fluorescent microscopy showed that blood vessels induced by FNIP1

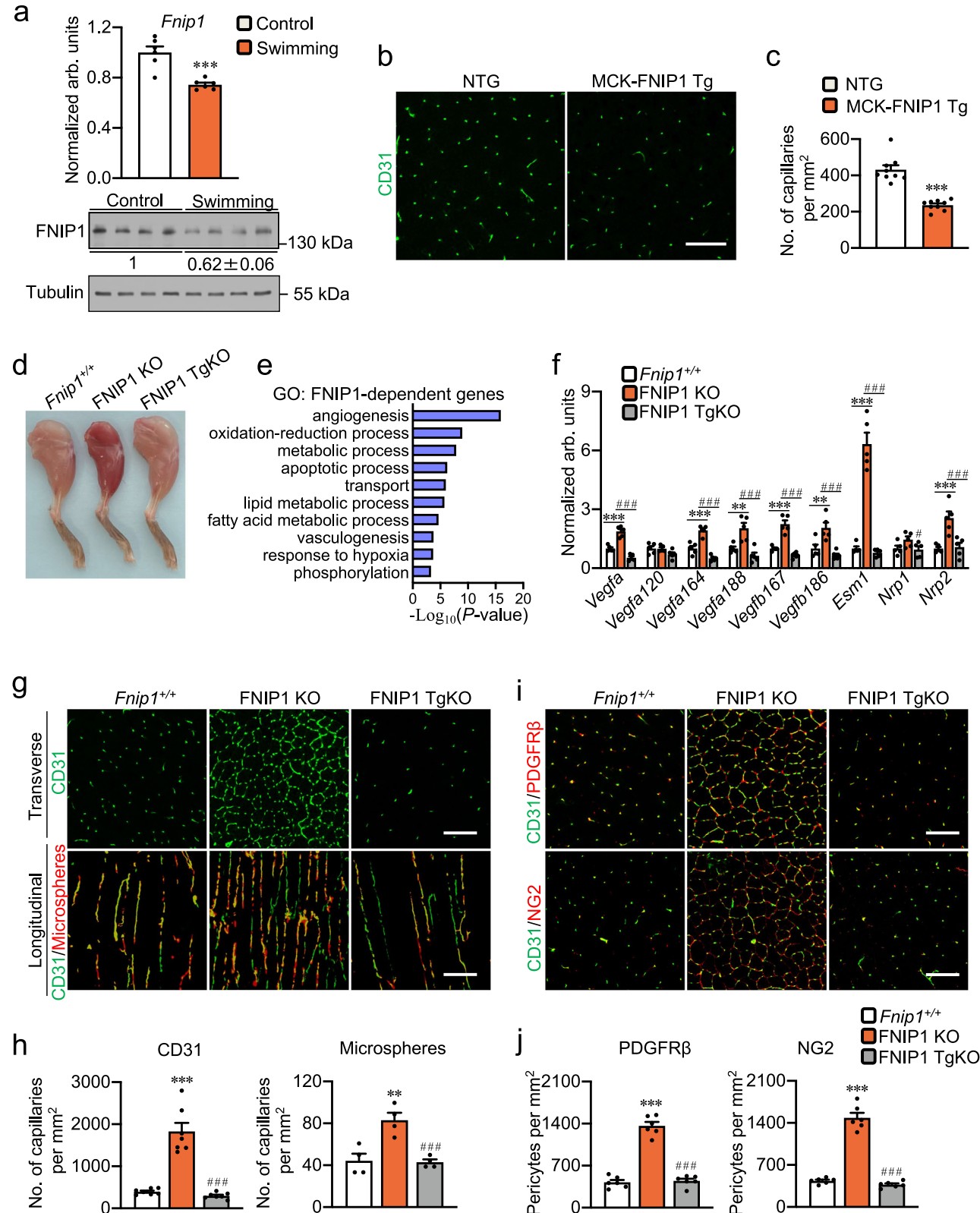

abrogation did not leak macromolecules EB, despite the dramatic increase in vessel density (Supplementary Fig. 2e). In contrast, as a positive control, a mouse model of muscle dystrophy (mdx mouse), in which EB is known to extravasate, had strong EB staining within myofibers (Supplementary Fig. 2e). Furthermore, vessel maturation requires the recruitment of perivascular cells, such as pericytes. Immunostaining of GC muscle cryosections for platelet-derived

growth factor (PDGF) receptor β, a pericyte-enriched marker, revealed an increase in the number of pericytes in FNIP1 KO mice but reduced in FNIP1 TgKO mice (Fig. 1i, j). The increase in pericyte density was also confirmed by chondroitin sulfate proteoglycan 4 (CSPG4; also named as NG2) staining, a second pericyte marker (Fig. 1i, j). Together, these data strongly demonstrate that FNIP1 is a crucial negative regulator for skeletal muscle angiogenesis.

**Fig. 1 | FNIP1-dependent regulation of skeletal muscle angiogenesis. a** Gene expression of *Fnip1* (qRT-PCR) in gastrocnemius (GC) muscles from indicated mice. *n* = 6 biologically independent mice. *P* value: 0.0005. Western blot analysis of FNIP1 from 14-week-old indicated mice. Quantification of FNIP1/Tubulin signal ratios were normalized (=1.0) to controls and presented below the corresponding bands. *n* = 4 biologically independent mice. **b, c** Representative images (**b**) and quantification (**c**) of CD31 immunofluorescent in GC muscles of 8-week-old mice. Scale bar, 100 μm. *n* = 9 biologically independent mice. *P* value: <0.0001. **d** Representative hindlimbs from 8-week-old mice. **e** Gene ontology (GO) enrichment analysis of gene transcripts regulated only in FNIP1 KO but not in FNIP1 TgKO muscles. The regulated pathways were reviewed using "Biological Process_Direct" term defined by GO, which is ranked by *P* value (one-sided Fisher's exact test). **f** Expression of genes (qRT-PCR) associated with angiogenesis in GC muscles from 8-week-old mice. *n* = 5 biologically independent mice. *P* value: <0.0001, <0.0001, 0.0026,

<0.0001, 0.0026, <0.0001, 0.0006. #*P* value: < 0.0001, <0.0001, 0.0002, <0.0001, 0.0002, <0.0001, 0.0458, 0.001. **g, h** Representative images (**g**) and quantification (**h**) of CD31 immunofluorescent and microsphere perfused images in GC muscles of 8-week-old mice. Scale bar, 100 μm. CD31, *n* = 7 biologically independent mice; microsphere, *n* = 4 biologically independent mice. *P* value: < 0.0001, 0.0012. #*P* value: < 0.0001, 0.0009. **i** Representative confocal images of CD31 (green) and PDGFRβ (red) co-staining, CD31 (green) and NG2 (red) co-staining in GC muscles. Scale bar, 100 μm. *n* = 6 biologically independent mice. **j** Quantification of pericytes per mm² in (**i**). *n* = 6 biologically independent mice. *P* value: <0.0001, <0.0001. #*P* value: <0.0001, <0.0001. All data are shown as the mean ± SEM. **P* < 0.01, ***P* < 0.001 vs. corresponding controls, ###*P* < 0.001 vs. FNIP1 KO, determined by two-tailed unpaired Student's *t* test (**a, c**), or one-way ANOVA (**f, h, j**) coupled to Fisher's least significant difference (LSD) post hoc test. Source data are provided as a Source Data file.

## Myofiber-specific ablation of FNIP1 promotes the formation of patent, functional blood vessels

To ascertain the role of myofiber FNIP1 in controlling muscle angiogenesis in vivo, we generated myofiber-specific FNIP1 KO mice (referred to as FNIP1 MKO) by crossing mice bearing a conditional *Fnip1* allele with introns 5 and 6 floxed (*Fnip1*^lox/lox) with the human skeletal actin promoter-driven Cre mice (*HSA-Cre*) (Supplementary Fig. 3a). FNIP1 MKO mice were born at normal Mendelian ratios and were grossly normal on inspection. As expected, the expression of *Fnip1* mRNA and protein levels were markedly reduced in skeletal muscles of FNIP1 MKO mice compared to *Fnip1*^f/f control littermates, whereas the protein levels of FNIP1 remain unchanged in liver and heart (Supplementary Fig. 3b, c). We further confirmed with a Rosa26mTmG cell fate-tracing mouse line that HSA-Cre activity was restricted to myofibers and was largely absent in non-muscle cells (Supplementary Fig. 3d).

Consistent with the observations in FNIP1 KO mice, myofiber-specific deletion of FNIP1 resulted in a pronounced change in the color of the skeletal muscle (Fig. 2a). We performed RNA-seq analysis on mRNA isolated from GC muscles of the FNIP1 MKO mice and littermate controls. We found that FNIP1 regulated a total of 3758 genes in skeletal muscles, with 1376 up- and 2382 downregulated, respectively (Fig. 2b). GO analysis of upregulated genes revealed significant enrichment in immune response, oxidative metabolism as well as angiogenesis (Fig. 2c). Gene expression validation studies demonstrated that the expression of many genes involved in angiogenesis was induced in the GC muscles of FNIP1 MKO mice compared to controls (Supplementary Fig. 3e). The induced expression of myoglobin protein paralleled the activation of angiogenesis in FNIP1 MKO muscles (Supplementary Fig. 3f). Consistent with the gene expression results, CD31 staining confirmed the marked increase of capillaries density in FNIP1 MKO muscle compared to *Fnip1*^f/f controls (Fig. 2d, e). The number of capillaries per mm² increased ~threefold (Fig. 2e). Because skeletal muscle angiogenesis involves the activation and proliferation of the EC, we also examined whether FNIP1 ablation affected EC in FNIP1 MKO mice. Using a fluorescence-activated cell sorting (FACS)-based method for quantitative analysis of the EC (CD31⁺, CD45⁻)[10], we observed a significant increase in the number of ECs from muscles of FNIP1 MKO mice (Supplementary Fig. 3g, h). Notably, no change in EC-specific marker gene expression was detected in ECs isolated from muscles of FNIP1 MKO mice compared to *Fnip1*^f/f controls (Supplementary Fig. 3i). This suggests that the EC cell niche was affected by myofiber FNIP1 deficiency. Moreover, examination of perfused microspheres in both GC and WV muscles revealed a marked increase in the number of patent vessels in FNIP1 MKO muscle compared to *Fnip1*^f/f controls (Fig. 2d, e and Supplementary Fig. 3j, k). EB dye infiltration test showed that blood vessels induced in FNIP1 MKO muscle did not leak macromolecules EB (Supplementary Fig. 3l). Furthermore, muscle cross sections were also stained for PDGFRβ. There was a positive correlation between capillary (CD31) and pericyte

(PDGFRβ) densities and pericyte coverage of endothelial cells was unchanged (Fig. 2f–h). Similar results were obtained when we conducted CD31 and NG2 co-staining (Fig. 2i–k). These data support that blood vessels induced by myofiber FNIP1 deletion were covered by pericytes. We further measured the blood flow in FNIP1 MKO muscles. Laser Doppler measurements showed that the blood flow to the skeletal muscles is higher in FNIP1 MKO muscles compared to *Fnip1*^f/f controls (Fig. 2l, m). Together, these data demonstrate that loss of FNIP1 in myofiber promotes the formation of patent, functional, and pericyte-covered blood vessels in skeletal muscle. The fact that two independent mouse models lacking myofiber FNIP1 show similar increased muscle angiogenesis phenotype strongly suggests that myofiber FNIP1 is a negative regulator for proximal vascular development in skeletal muscle.

## AMPK-independent regulation of muscle angiogenesis by FNIP1

We have recently uncovered that FNIP1 regulates muscle mitochondrial oxidative function through metabolic regulator AMPK. Given the known role of AMPK in tissue vascular remodeling[16,19], we thus sought to determine whether myofiber FNIP1 mediates its effect on muscle angiogenesis program through AMPK. We examined the skeletal muscles of the *Fnip1*^−/−, AMPKα1/α2^f/f/Myf5-Cre (TKO) mice[26]. Interestingly, we found that muscle-specific disruption of AMPKα1/α2 did not change the red coloration in FNIP1 KO muscles (Fig. 3a). To more thoroughly analyze pathways that are affected by AMPKα1/α2 ablation in the context of FNIP1 KO, we performed RNA-seq transcriptome analysis of GC muscles from *Fnip1*^+/+, FNIP1 KO and TKO mice. Surprisingly, whereas we found that total AMPKα1/α2 loss blunted a subset of FNIP1-regulated genes in FNIP1 KO muscles (Fig. 3b), the majority of differentially regulated genes in FNIP1 KO muscles were not affected by AMPKα1/α2 ablation (Fig. 3b). GO analysis of AMPK-independent genes revealed significant enrichment in angiogenesis processes (Fig. 3c). Gene expression validation studies demonstrated that many proangiogenic genes were induced in both FNIP1 KO and TKO muscles (Fig. 3d). We also examine the expression of antiangiogenic genes in FNIP1 KO muscles, while the expression of many antiangiogenic genes (*Ppargc1b*, *Thbs1*, *Thbs2*, *Vash1*, and *Endo1*) was not different in the FNIP1 KO compared with the *Fnip1*^+/+ muscles, *Pedf* and *Plasminogen* mRNA levels were decreased in FNIP1 KO muscles (Supplementary Fig. 4a). Interestingly, total AMPKα1/α2 loss reversed the *Plasminogen*, but not *Pedf*, expression in FNIP1 KO muscles (Supplementary Fig. 4a). No significant changes in CD31 staining were observed in GC muscles of TKO mice compared to FNIP1 KO mice (Fig. 3e, f). Moreover, microspheres perfusion studies demonstrate that muscle functional vascularity was increased in GC and WV muscles of FNIP1 KO mice but not affected by AMPKα1/α2 disruption (Fig. 3e, g and Supplementary Fig. 4b, c). Furthermore, PDGFRβ and NG2 staining also showed an increase in the number of pericytes in GC muscles of TKO mice compared to *Fnip1*^+/+ mice (Fig. 3f, g). Together, these data

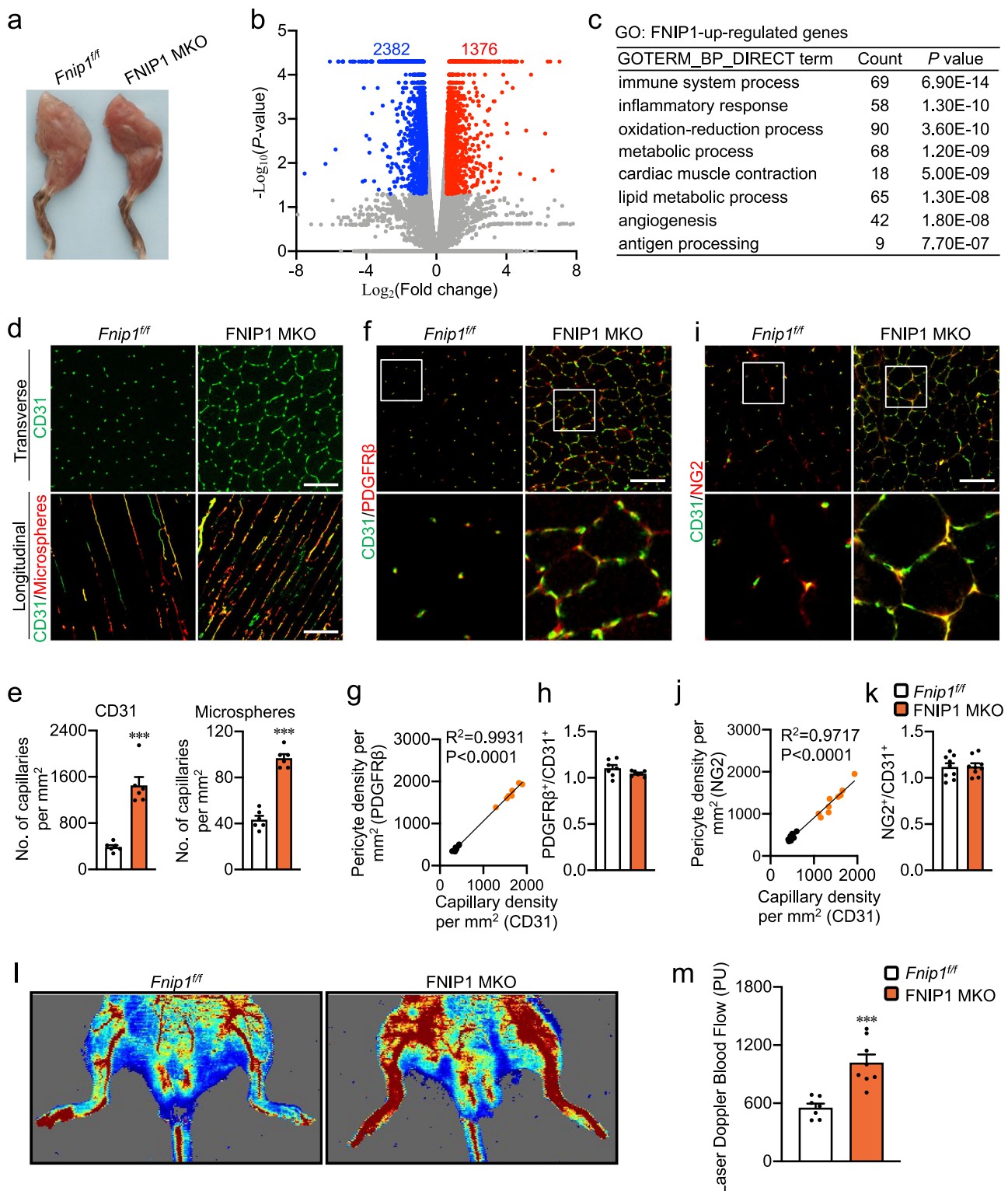

## Myofiber FNIP1 regulates muscle angiogenesis through macrophage recruitment

How might myofiber FNIP1 regulate proximal vascular development? Evidence is emerging that macrophages are a crucial component of skeletal muscle angiogenesis remodeling[7,8,12]. In fact, the gene sets most strongly induced in FNIP1 MKO muscles were those of the immune system and inflammatory response (Fig. 2c), suggesting that

suggest that AMPK is dispensable for the FNIP1-mediated control of muscle angiogenesis.

myofiber FNIP1 deficiency influences immune cell infiltration in skeletal muscle. We next study whether loss of myofiber FNIP1 affects macrophage recruitment in skeletal muscle. To test this, sections from GC muscles of FNIP1 MKO mice were immunostained with antibodies to the macrophage-specific marker F4/80 (Fig. 4a). This revealed a dramatic -threefold increase in the number of macrophages in the FNIP1 MKO muscle when compared with Fnip1[f/f] controls (Fig. 4a). Real-time quantitative PCR analysis of RNA from the GC muscles further confirmed the induction of many genes known to be expressed strongly in M1 or M2 macrophages (Fig. 4b). FNIP1 MKO muscle

**Fig. 2 | Myofiber-specific ablation of FNIP1 promotes the formation of patent, functional blood vessels. a** Representative hindlimbs from *Fnip1f/f* and FNIP1 MKO mice at the age of 8 weeks. **b** Volcano plot showing fold changes versus *P* values for analyzed RNA-seq data generated from the GC muscles of 8-week-old male FNIP1 MKO mice compared with *Fnip1f/f* littermate controls. Significantly upregulated genes are represented by red dots, whereas downregulated genes are represented by blue dots. **c** GO enrichment analysis ("Biological Process_Direct" term) of gene transcripts upregulated in FNIP1 MKO muscle. **d** Representative images of CD31 immunofluorescent (green) and microsphere perfused (red) images in GC muscles from 14-week-old *Fnip1f/f* and FNIP1 MKO mice. Scale bar, 100 μm. **e** Quantification of capillaries per mm² in (**d**). *n* = 6 biologically independent mice per group. *P* value: 0.0001, <0.0001. **f** Representative confocal images of immunostainings of CD31 (green) and PDGFRβ (red) co-staining in *Fnip1f/f* and FNIP1 MKO GC muscles. Scale bar, 100 μm. **g** Pearson's correlation of pericyte density (PDGFRβ) and Capillary

density (CD31). *P* value: <0.0001. **h** The ratio of PDGFRβ⁺/CD31⁺ in *Fnip1f/f* and FNIP1 MKO GC muscles. *n* = 7 biologically independent mice per group. **i** Representative confocal images of immunostainings of CD31 (green) and NG2 (red) co-staining in *Fnip1f/f* and FNIP1 MKO GC muscles. Scale bar, 100 μm. **j** Pearson's correlation of pericyte density (NG2) and Capillary density (CD31). *P* value: <0.0001. **k** The ratio of NG2⁺/CD31⁺ in *Fnip1f/f* and FNIP1 MKO GC muscles. *n* = 9 biologically independent mice per group. **l** Representative laser Doppler images of 14-week-old *Fnip1f/f* and FNIP1 MKO mice. **m** Quantification of hindlimb microcirculation blood perfusion, in perfusion unit (PU). *Fnip1f/f*, *n* = 7; FNIP1 MKO, *n* = 8 biologically independent mice. *P* value: 0.0004. All data are shown as the mean ± SEM. ***P* < 0.001 vs. corresponding *Fnip1f/f* controls, determined by two-tailed unpaired Student's *t* test, except in (**g**, **j**) where two-tailed Pearson correlation was used. Source data are provided as a Source Data file.

cryosections stained for the M1 macrophage marker CD80 or M2 macrophage marker CD206 showed an increase in both M1 and M2 macrophage recruitment compared with the *Fnip1f/f* control muscles (Fig. 4c, d). These observations suggested that loss of myofiber FNIP1 leads to infiltration of macrophages in skeletal muscle. Moreover, we also confirmed the strong recruitment of macrophages in FNIP1 KO but not in FNIP1 TgKO muscles (Supplementary Fig. 5a, b). In addition, no significant changes in macrophage recruitment were observed in the muscles of TKO mice compared to FNIP1 KO mice (Supplementary Fig. 5c), which is consistent with AMPK-independent regulation of muscle angiogenesis by FNIP1.

To assess the requisite role of FNIP1 in the control of adult muscle angiogenesis, and to determine whether macrophage recruitment is required in this mechanism, adeno-associated virus (AAV9) Cre-mediated FNIP1 deletion in adult muscle were conducted in *Fnip1f/f* mice. The effects of liposome-encapsulated clodronate, a chemical chaperone known to induce macrophages apoptosis, were assessed in the presence of AAV9-Cre-mediated FNIP1 ablation (Fig. 4e, h). Compared to *Fnip1f/f* muscle injected with control AAV9-GFP viruses, AAV9-Cre-mediated deletion of FNIP1 in skeletal muscle resulted in lower levels of *Fnip1* mRNA (Fig. 4e and Supplementary Fig. 5d). Consistent with the results from FNIP1 MKO mice, AAV9-based FNIP1 ablation in *Fnip1f/f* muscles also leads to activated angiogenic program marked by red muscle color, induction of proangiogenic factors, and angiogenesis (Fig. 4f and Supplementary Fig. 5e–g). Indicative of activated macrophage recruitment, the F4/80 staining signals were significantly increased in muscles from *Fnip1f/f* mice injected with AAV9-Cre compared to control viruses (Fig. 4f). The activation of macrophage infiltration was further confirmed by the increased expression of macrophage marker genes in muscles lacking FNIP1 (Fig. 4g). Therefore, loss of FNIP1 in adult muscles also drives macrophages recruitment and muscle angiogenesis. As expected, clodronate treatment resulted in marked reduction of macrophages recruitment in the muscles from *Fnip1f/f* mice injected with AAV9-Cre (Fig. 4h–k). FNIP1 deficiency-mediated increased CD31 staining was significantly reduced in the presence of clodronate (Fig. 4j, k). Moreover, clodronate treatment also resulted in marked reduction of the number of pericytes in muscles from *Fnip1f/f* mice injected with AAV9-Cre (Fig. 4j, k). We also explore the potential macrophage-mediated EC cell niche activation. Ki67 staining showed that AAV9-based FNIP1 ablation in *Fnip1f/f* muscles leads to a dramatic induction of EC proliferation, clodronate treatment resulted in marked reduction of EC proliferation in the muscles from *Fnip1f/f* mice injected with AAV9-Cre (Supplementary Fig. 5h, i). Moreover, TUNEL staining analysis showed that FNIP1 deficiency-mediated decreased EC cell apoptosis was also prevented in the presence of clodronate (Supplementary Fig. 5j, k). These results further establish the relevance of FNIP1-mediated regulation of muscle angiogenesis and demonstrate the importance of macrophages recruitment in this mechanism.

## FNIP1-dependent regulation of macrophage recruitment and muscle angiogenesis by the transcriptional coactivator PGC-1α

To define the mechanism involved in the regulation of macrophages recruitment by myofiber FNIP1, the gene expression profiling data sets of FNIP1 MKO muscle were further analyzed. Comparative analysis identified that genes involved in cytokine and chemokine signaling pathways were upregulated in FNIP1-deficient muscles (Fig. 5a and Supplementary Fig. 6a). Of particular interest was the expression pattern of many chemokine genes. Many genes known to recruit macrophages (*Ccl5*, *Ccl8*, *Ccl12*, *Cxcl9*, *Cxcl10*, *Cxcl13*, etc.) were upregulated in FNIP1 MKO muscles (Fig. 5b), a pattern that was also observed for FNIP1 KO but not FNIP1 TgKO (Supplementary Fig. 6b). These findings were validated by quantitative RT-PCR (Fig. 5c and Supplementary Fig. 6c).

Previous studies have suggested a role of PGC-1α in regulating myocyte-secreted factors in skeletal muscle[14,37]. Indeed, we found that *Ppargc1a* mRNA expression levels were induced in FNIP1 MKO muscles (Fig. 5d). Western blotting further confirmed that the marked increased expression of PGC-1α protein in FNIP1 MKO muscles (Fig. 5e). PGC-1α is known to regulate muscle gene expression by co-activating many nuclear receptor family members[38,39]. The chemokine *Ccl8* and *Cxcl13* gene promoter regions were thus screened for putative nuclear receptor binding sties. We identified several potential nuclear receptor recognition half site (AGGTCA) in the *Ccl8* and *Cxcl13* cis-promoter regions (Fig. 5f). Cell co-transfection studies were next conducted using a mouse 4.7 kb *Cxcl13* promoter reporter containing the nuclear receptor recognition element. The *Cxcl13* luciferase reporter was not activated by PGC-1α or ERRβ alone, but when expressed together, synergistic activation was observed (Fig. 5g). Taken together, these findings indicate that myofiber FNIP1 deficiency induces PGC-1α to activate chemokine genes transcription.

We next asked whether knocking out PGC-1α would be sufficient to ablate macrophage recruitment and angiogenesis program in muscles from FNIP1 KO mice. We took advantage of the *Fnip1⁻/⁻*, PGC-1αf/f/MCK-Cre (DKO) mice, in which the PGC-1α gene is disrupted in muscles in FNIP1 KO background[26]. We confirmed that the induced expression of *Ppargc1a* mRNA was abolished in the DKO muscles (Fig. 6a). Muscle-specific disruption of PGC-1α resulted in marked reduction of macrophages recruitment in the muscles from FNIP1 KO mice (Fig. 6b). FNIP1 deficiency-mediated induction of macrophage genes was significantly reduced in the absence of PGC-1α (Fig. 6c). Moreover, deletion of PGC-1α also abolished the induction of chemokine genes expression in FNIP1 KO muscles (Fig. 6d). These data suggest that PGC-1α is required for the activated chemokine gene expression and macrophage recruitment in FNIP1-deficient muscles.

A series of studies were next conducted to determine the role of PGC-1α in FNIP1-mediated skeletal muscle angiogenesis. As shown in Fig. 6e, muscle-specific disruption of PGC-1α abolished the red coloration in FNIP1 KO muscles (Fig. 6e). We also performed RNA-seq transcriptome analysis of skeletal muscles from *Fnip1+/+*, FNIP1 KO, and

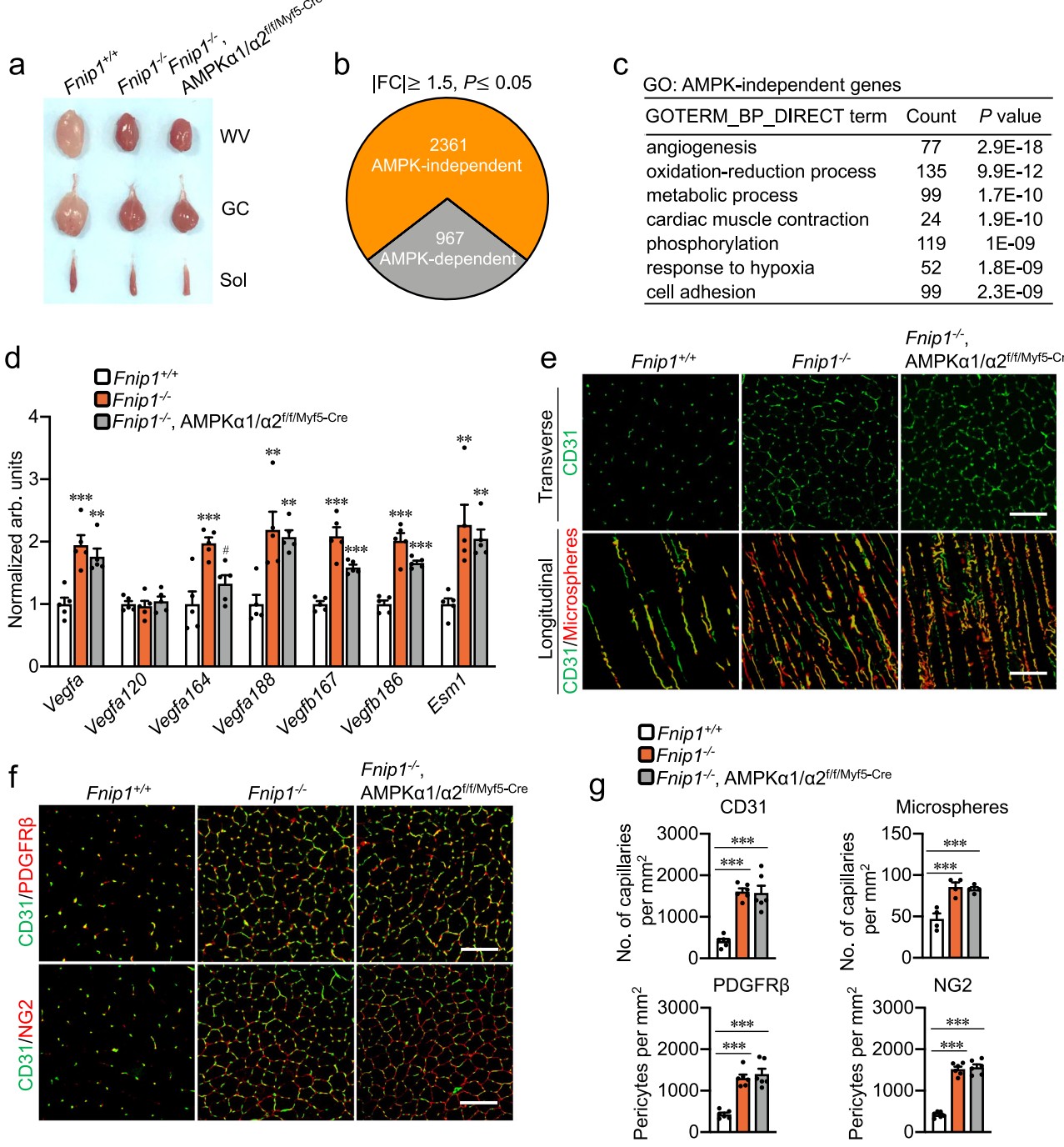

**Fig. 3 | AMPK-independent regulation of muscle angiogenesis by FNIP1.**
**a** Representative white vastus lateralis (WV), GC and soleus muscles from indicated mice at the age of 8 weeks. **b** Schematic of identification of AMPK-dependent and -independent genes regulated by FNIP1 with the cutoff criteria of a fold change greater than 1.5 (either direction) and a cutoff significant level of $P < 0.05$. **c** GO enrichment analysis ("Biological Process_Direct" term) of AMPK-independent gene transcripts regulated by FNIP1. **d** Expression of genes (qRT-PCR) associated with angiogenesis in GC muscles from the indicated genotypes. $n = 5$ biologically independent mice per group. *$P$ value: 0.0003, 0.0016, 0.0007, 0.0012, 0.0024, <0.0001, 0.0007, <0.0001, 0.0001, 0.0011, 0.0043. #$P$ value: 0.0106.
**e** Representative images of CD31 immunofluorescent (green) and microsphere perfused (red) images in GC muscles from 8-week-old $Fnip1^{+/+}$, $Fnip1^{-/-}$ and $Fnip1^{-/-}$, AMPKα1/α2$^{f/f/Myf5-Cre}$ mice. Scale bar, 100 μm. $n = 6$ (CD31) and $n = 4$ (microsphere) biologically independent mice per group. **f** Representative images of immunostainings of CD31 (green) and PDGFRβ (red) co-staining, CD31 (green) and NG2 (red) co-staining in GC muscles from indicated mice. Scale bar, 100 μm. $n = 6$ biologically independent mice per group. **g** Quantification of capillaries and pericytes per mm² in (**e**, **f**). $n = 6$ biologically independent mice per group, $P$ value: <0.0001, <0.0001, 0.0004, 0.0007, <0.0001, <0.0001, <0.0001, <0.0001. All data are shown as the mean ± SEM. **$P < 0.01$, ***$P < 0.001$ vs. corresponding $Fnip1^{+/+}$ controls, #$P < 0.05$ vs. $Fnip1^{-/-}$, determined by one-way ANOVA coupled to Fisher's least significant difference (LSD) post hoc test. Source data are provided as a Source Data file.

DKO mice to gain insight into the PGC-1α-dependent regulation of skeletal muscle angiogenesis. GO analysis of PGC-1α-dependent genes revealed significant enrichment in angiogenesis processes (Fig. 6f, g). Gene expression validation studies confirmed that FNIP1 deficiency-

mediated induction of angiogenesis marker genes were significantly reduced in the absence of PGC-1α (Supplementary Fig. 6d). Moreover, muscle-specific disruption of PGC-1α resulted in a marked decrease in CD31 staining in the muscles of FNIP1 KO mice (Fig. 6h, i). Furthermore,

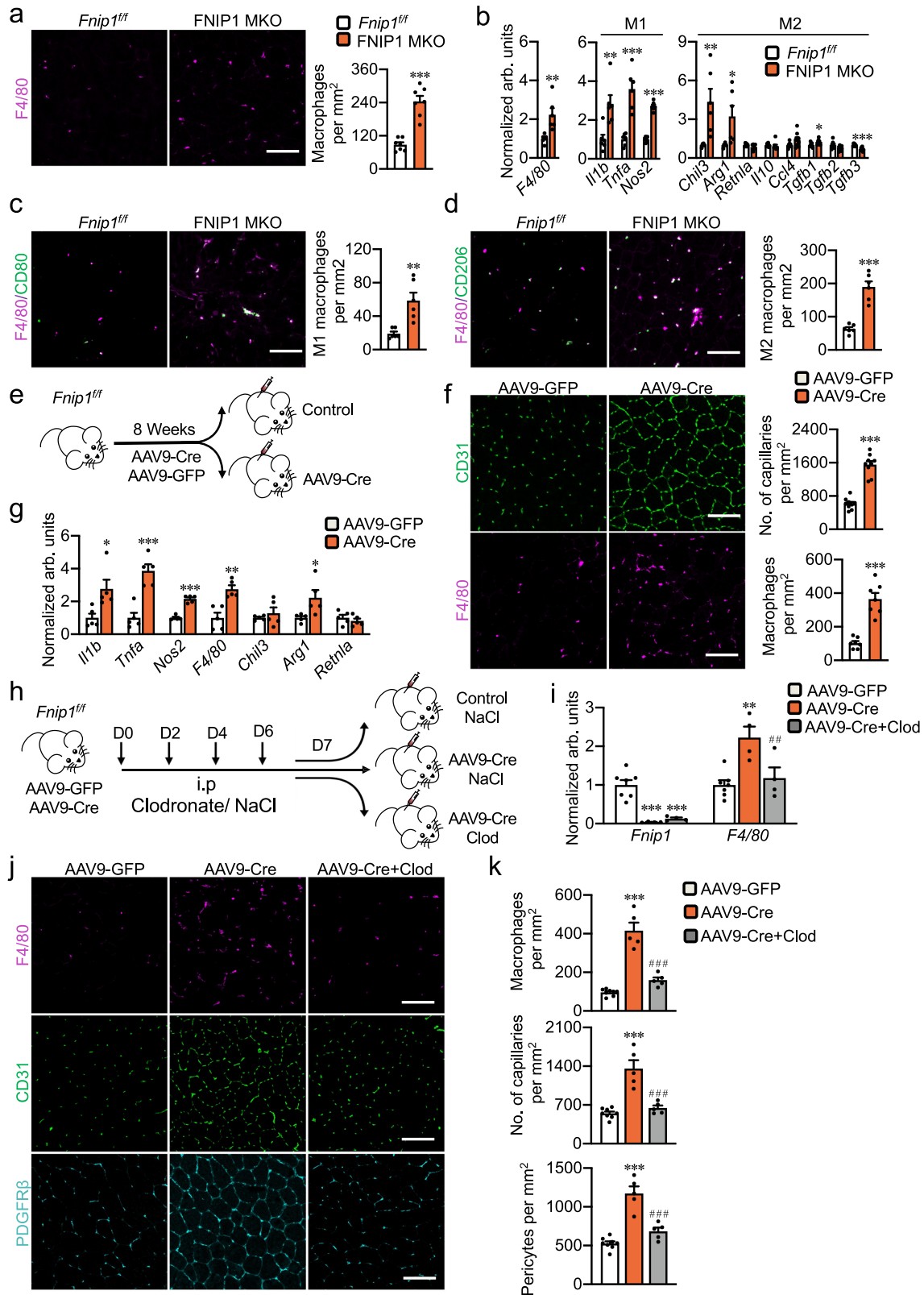

microspheres perfusion studies demonstrate that FNIP1 deficiency-mediated increase in the number of patent vessels was significantly abolished by PGC-1α disruption across several muscle types (Fig. 6h, i and Supplementary Fig. 6e–g). In addition, the deletion of PGC-1α also resulted in marked reduction of the number of pericytes in muscles from FNIP1 KO mice (Fig. 6j, k). We also observed PGC-1α-dependent regulation of muscle angiogenesis in female FNIP1 KO mice

(Supplementary Fig. 6h, i). Together, these data suggest that FNIP1 regulates muscle functional angiogenesis through PGC-1α-dependent macrophage recruitment.

Previous studies have revealed that FNIP1 could act as co-chaperone that regulates the chaperone function of Hsp90[29,30,40,41]. We also sought to determine whether the chaperone function of Hsp90 is involved in the regulation of PGC-1α, we found that Hsp90

**Fig. 4 | Myofiber FNIP1 regulates muscle angiogenesis through macrophage recruitment. a** Representative images and quantification of F4/80 immunofluorescent staining in GC muscles from 14-week-old mice. Scale bar, 100 μm. $n = 7$ biologically independent mice. *P* value: <0.0001. **b** Expression of macrophage activation genes (qRT-PCR) in GC muscles. $n = 8$ biologically independent mice. *P* value: 0.0049, 0.0047, 0.0001, <0.0001, 0.008, 0.026, 0.0165, 0.0002. **c** Representative images and quantification of M1 macrophage (CD80) (green) and F4/80 (magenta) co-staining in 8-week-old mice GC muscles. $n = 6$ biologically independent mice. Scale bar, 100 μm. *P* value: 0.0024. **d** Representative images and quantification of M2 macrophage (CD206) (green) and F4/80 (magenta) co-staining in GC muscles. $n = 6$ biologically independent mice. Scale bar, 100 μm. *P* value: <0.0001. **e** Schematic showing adeno-associated virus (AAV9) Cre-mediated FNIP1 deletion in muscle. **f** Representative images and quantification of F4/80 and CD31 immunofluorescent staining in muscles from Fnip1$^{f/f}$ mice injected with AAV9-Cre or control viruses. Scale bar, 100 μm. $n = 7$ (F4/80) and $n = 9$ (CD31)

biologically independent mice. *P* value: <0.0001, <0.0001. **g** Expression of macrophage activation genes in muscles. $n = 5$ biologically independent mice. *P* value: 0.0214, 0.0005, <0.0001, 0.0021, 0.0342. **h** Schematic illustrating the liposome-encapsulated clodronate treatment in the presence of AAV9-Cre-mediated FNIP1 ablation in 8-week-old mice. **i** qRT-PCR analysis of mRNA levels in muscles. AAV-GFP, $n = 7$, AAV-Cre and AAV-Cre+Clod, $n = 4$ biologically independent mice. *P* value: <0.0001, <0.0001, 0.0011. #*P* value: 0.007. **j, k** Representative images (**j**) and quantification (**k**) of F4/80, CD31 and PDGFRβ immunofluorescent staining in muscles. Scale bar, 100 μm. AAV-GFP, $n = 8$, AAV-Cre and AAV-Cre+Clod, $n = 5$ biologically independent mice. *P* value: <0.0001, <0.0001, <0.0001. #*P* value: <0.0001, <0.0001, <0.0001. All data are shown as the mean ± SEM. *$P < 0.05$, **$P < 0.01$, ***$P < 0.001$ vs. corresponding controls, ##$P < 0.01$, ###$P < 0.001$ vs. AAV9-Cre, determined by two-tailed unpaired Student's *t* test (**a**–**d**, **f**, **g**) or one-way ANOVA (**i**, **k**) coupled to Fisher's least significant difference (LSD) post hoc test. Source data are provided as a Source Data file.

protein levels were not altered in FNIP1 MKO muscle (Supplementary Fig. 7a). In addition, Hsp90 inhibition by 17DMAG treatment or siRNAs in C2C12 myocytes had no effect on *Ppargc1a* mRNA levels (Supplementary Fig. 7b, c). Vascular leakiness is regulated by mechanisms that control endothelial barrier function, and previous studies have identified the main component of endothelial junctions, VE-cadherin, that are crucial for endothelial barrier function[42]. Whereas the expression of many VE-cadherin genes was not changed in FNIP1 KO muscles versus controls, we found that VE-cadherin genes *Cdh5* and *Cdh20* mRNA levels were significantly increased in FNIP1 KO muscles (Supplementary Fig. 7d, e). These results were of interest because the CDH5 has been shown to be critical for the maintenance of endothelial barrier[42–44]. Western blotting further confirmed that the increased expression of CDH5 protein in FNIP1 MKO muscles (Supplementary Fig. 7f). Moreover, muscle-specific deletion of PGC-1α abolished the induction of CDH5 protein expression in FNIP1 KO muscles (Supplementary Fig. 7g). Together, these results suggest that myofiber FNIP1/PGC-1α signaling may induce CDH5 gene expression to improve endothelial barrier function. Interestingly, gene expression studies demonstrated that the expression of the *Vegf* genes was induced by FNIP1 deficiency but not affected by the clodronate treatment (Supplementary Fig. 8a), suggesting factors from outside the VEGF family could be involved in the activation of angiogenesis by FNIP1 deficiency. Indeed, comparative analysis showed that many angiogenesis factors were induced by FNIP1 deficiency but not affected by the total loss of AMPKα1/α2 in muscles (Supplementary Fig. 8b). Particularly, the *Tnfa* and *Nos2*, which are induced in FNIP1 KO muscles, have been shown to regulate vascular niche and angiogenesis[45–48]. RT-qPCR confirmed that both *Tnfa* and *Nos2* gene expression were significantly induced in FNIP1 KO muscles but not affected by the total loss of AMPKα1/α2 (Supplementary Fig. 8c). However, gene expression validation studies revealed that FNIP1 deficiency-mediated increased *Tnfa* and *Nos2* expression was reduced after clodronate treatment (Supplementary Fig. 8d). Moreover, we also confirmed the induction of *Tnfa* and *Nos2* expression was abolished in FNIP1 TgKO muscles (Supplementary Fig. 5b), and muscle-specific disruption of PGC-1α resulted in marked decreased in *Tnfa* and *Nos2* expression in the muscles of FNIP1 KO mice (Fig. 6c). Together, these data show the strong correlation between *Tnfa* and *Nos2* expression and muscle angiogenesis in our series of mouse model studies.

### Deletion of myofiber FNIP1 enhances hindlimb ischemia (HLI)-induced revascularization

To address the translational implication of myofiber FNIP1-dependent control of muscle angiogenesis, we asked whether myofiber FNIP1 might regulate ischemic muscle revascularization or neoangiogenesis. We focused on a preclinical model for peripheral vascular disease, in which unilateral hindlimb ischemia is surgically induced in mice by ligating the femoral vessels in the left hindlimb, whereas the right

hindlimb serves as a nonischemic control[49]. The ischemic hindlimb undergoes robust revascularization over time and serves as a good model for exploring the regulation of neoangiogenesis[49]. To determine the role of FNIP1 in ischemic revascularization, we applied unilateral hindlimb ischemia to both FNIP1 MKO and *Fnip1*$^{f/f}$ littermate mice, and we assessed macrophage recruitment and neoangiogenesis after the induction of hindlimb ischemia in both the contralateral and ischemic muscles (Fig. 7a–f). The ischemic FNIP1 MKO muscle cryosections stained for the macrophage marker F4/80 showed a more dramatic increase in macrophage recruitment compared with the ischemic *Fnip1*$^{f/f}$ muscle (Fig. 7a). Ischemia led to greater induction of both M1 and M2-macrophages marker genes expression in muscle from FNIP1 MKO mice (Fig. 7b). Moreover, CD31, NG2 and PDGFRβ staining revealed accelerated revascularization in FNIP1 MKO compared to *Fnip1*$^{f/f}$ ischemic muscles (Fig. 7c–f). Together, these data suggest that myofiber FNIP1 inhibits ischemic neoangiogenesis.

We also monitored skeletal muscle revascularization from ischemia using noninvasive laser Doppler imaging. The day of surgery was considered as day 0. Laser Doppler imaging showed that blood flow to the ischemic hindlimb upon HLI was about 90% lower than that in contralateral nonischemic hindlimb in both *Fnip1*$^{f/f}$ and FNIP1 MKO mice (Fig. 7g, h), confirming that hindlimb vascular occlusion has been equally applied to both genotypes. In *Fnip1*$^{f/f}$ mice, ischemia-induced muscle damage is followed by progressive neovascularization and return of blood flow to the limb over the ensuing weeks (Fig. 7g, h). Remarkably, in FNIP1 MKO mice, the return of blood flow after ligation of the femoral artery is markedly accelerated (Fig. 7g, h): 71% of blood flow is recaptured in these animals within 10 days after surgery, in contrast with less than 50% in *Fnip1*$^{f/f}$ controls. Therefore, loss of myofiber FNIP1 actually accelerates ischemic skeletal muscle revascularization. Collectively, these findings indicate that myofiber FNIP1 limits ischemic muscle neoangiogenesis, inactivation of which can enhance the process of revascularization.

## Discussion

Ischemia of the heart and limbs attributable to compromised blood supply is a major cause of mortality and morbidity worldwide. Delineation of the molecular mechanisms involved in regulating functional angiogenesis and revascularization from ischemia is of great importance to the development of therapeutic approaches for ischemic diseases. In this study, we uncovered a crucial role for myofiber FNIP1 in the control of skeletal muscle functional angiogenesis and the governance of muscle revascularization from ischemia (Fig. 8). We showed that FNIP1 expression is downregulated in response to exercise training in skeletal muscle. Mice overexpressing FNIP1 in myofiber have limited muscle angiogenesis, whereas myofiber-specific deletion of FNIP1 markedly promotes the formation of patent, functional blood vessels. Mechanistically, the increased muscle angiogenesis phenotype is not due to AMPK activation but greater macrophage recruitment in

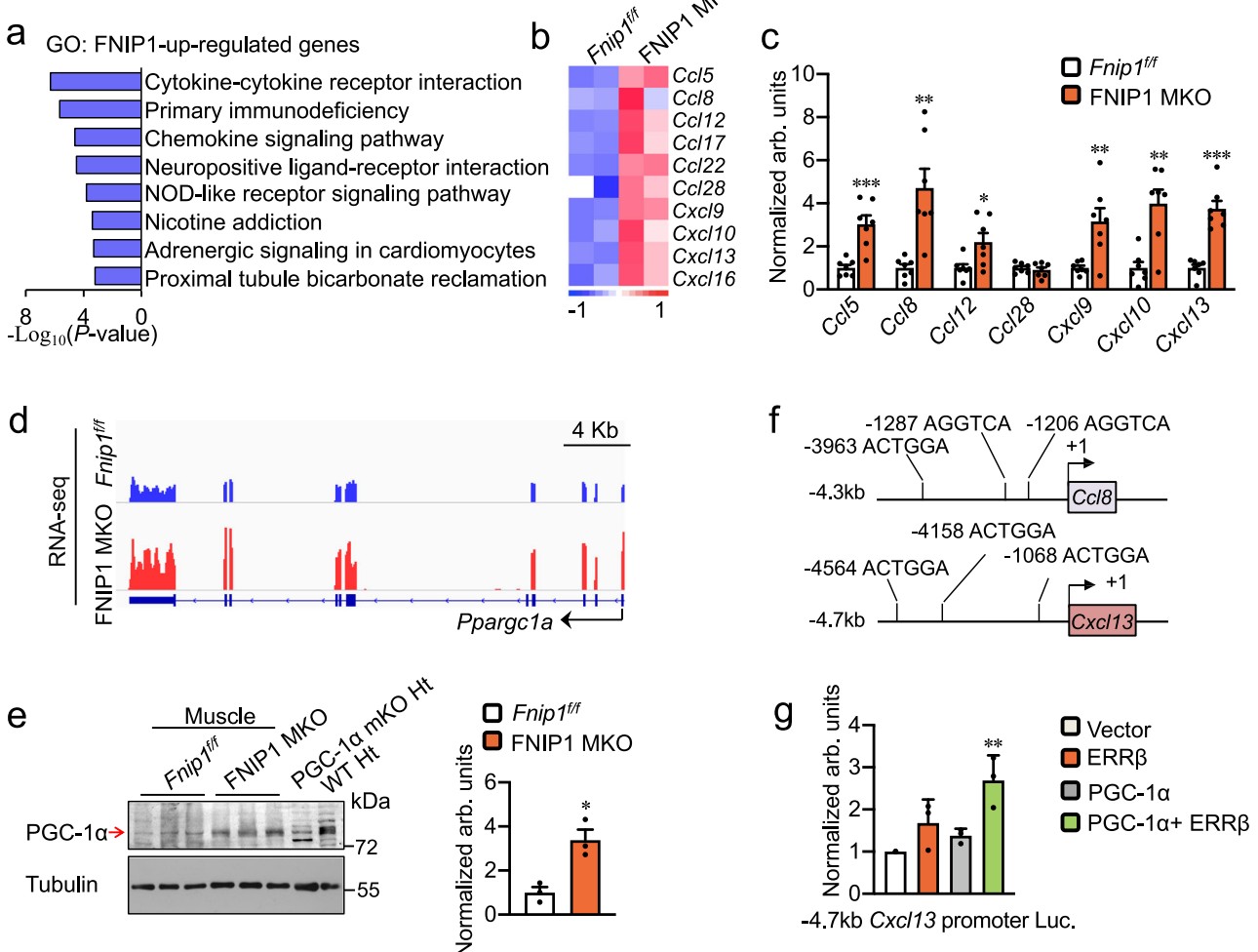

**Fig. 5 | Muscle FNIP1 deficiency induces PGC-1α to activate chemokine gene expression and macrophage recruitment. a** GO enrichment analysis (KEGG pathway) of gene transcripts upregulated in FNIP1 MKO muscle, with the top eight terms shown, which is ranked by *P* value (one-sided Fisher's exact test). **b** Heatmap analysis of chemokine genes regulated in FNIP1 MKO muscle compared with *Fnip1^{f/f}* controls. *n* = 2 independent samples per group. Color scheme for fold change is provided. **c** Expression of genes (qRT-PCR) involved in Chemokine signaling pathway in GC muscles from the 14-week-old indicated genotypes. *n* = 7 biologically independent mice per group. *P* value: 0.0004, 0.0015, 0.0215, 0.0052, 0.0012, <0.0001. **d** Genome browser tracks of RNA-seq data were visualized in IGV. **e** Western blot analysis of PGC-1α from the indicated mice. Quantification of the PGC-1α/Tubulin signal ratios were normalized (=1.0) to *Fnip1^{f/f}* controls. *n* = 3

biologically independent mice per group. *P* value: 0.0117. **f** The schematics show the location of the putative conserved nuclear receptor recognition half site (AGGTCA) relative to the *Ccl8* and *Cxcl13* gene transcription start site (+1). **g** PGC-1α and ERRβ synergistically activate *Cxcl13* gene promoter. The mouse *Cxcl13*.Luc.4.7k promoter reporter was used in co-transfection studies in HEK293 cells in the presence of expression vectors indicated. Values represent mean (± SEM) firefly/renilla luciferase activity shown as arbitrary units (arb. units) normalized (=1.0) to vector control. *n* = 3 independent experiments. *P* value: 0.0012. All data are shown as the mean ± SEM. *$P < 0.05$, **$P < 0.01$, ***$P < 0.001$ vs. corresponding *Fnip1^{f/f}* or Vector controls, determined by two-tailed unpaired Student's *t* test (**c**, **e**), or one-way ANOVA (**g**) coupled to Fisher's least significant difference (LSD) post hoc test. Source data are provided as a Source Data file.

FNIP1-depleted muscles. Further, we uncovered a mechanism for the regulation of muscle macrophage recruitment mediated by the concerted action of chemokine genes that are activated by the FNIP1-PGC-1α axis in myofibers. Remarkably, in the mouse hindlimb ischemia model of peripheral artery disease, loss of myofiber FNIP1 resulted in improved recovery of blood flow. Taken together, our study establishes a pivotal role of FNIP1 as a negative regulator of functional angiogenesis in skeletal muscle. This FNIP1-macrophage regulatory circuit shows promise for the identification of therapeutic targets aimed at enhancing blood flow recovery in a variety of ischemic disease states such as peripheral artery disease and heart failure.

Previous studies have identified multiple regulatory pathways, such as nuclear receptor ERRs and HIF, along with coregulators PGC-1α in driving the skeletal muscle angiogenesis program[14–18]. However, the counterbalancing negative regulatory mechanisms remain less clear. Here, we identify FNIP1 acts as an endogenous "brake" that negatively

regulates functional vascularization in skeletal muscle. Interestingly, FNIP1 is downregulated in response to exercise training. Loss of myofiber FNIP1 promotes functional angiogenesis in skeletal muscle and likely recapitulates the complex physiological angiogenesis elicited by exercise. Therefore, myofiber FNIP1 may serve as a rheostat to prevent uncontrolled and aberrant angiogenesis, which needs to be removed during exercise.

Our results indicate that in addition to the known role in the regulation of mitochondrial function muscle fiber type[22,26], myofiber FNIP1 exerts control actions upon skeletal muscle angiogenesis program through macrophage recruitment, independent of AMPK. Thus, our study has expanded the role of FNIP1 to include pivotal functional angiogenesis beyond AMPK signaling. Interestingly, FNIP1 interacting partner FLCN has also been shown to regulate angiogenic gene expression in muscle[50]. Notably, slow-oxidative myofibers inherently express higher levels of angiogenic factors and recruit more blood

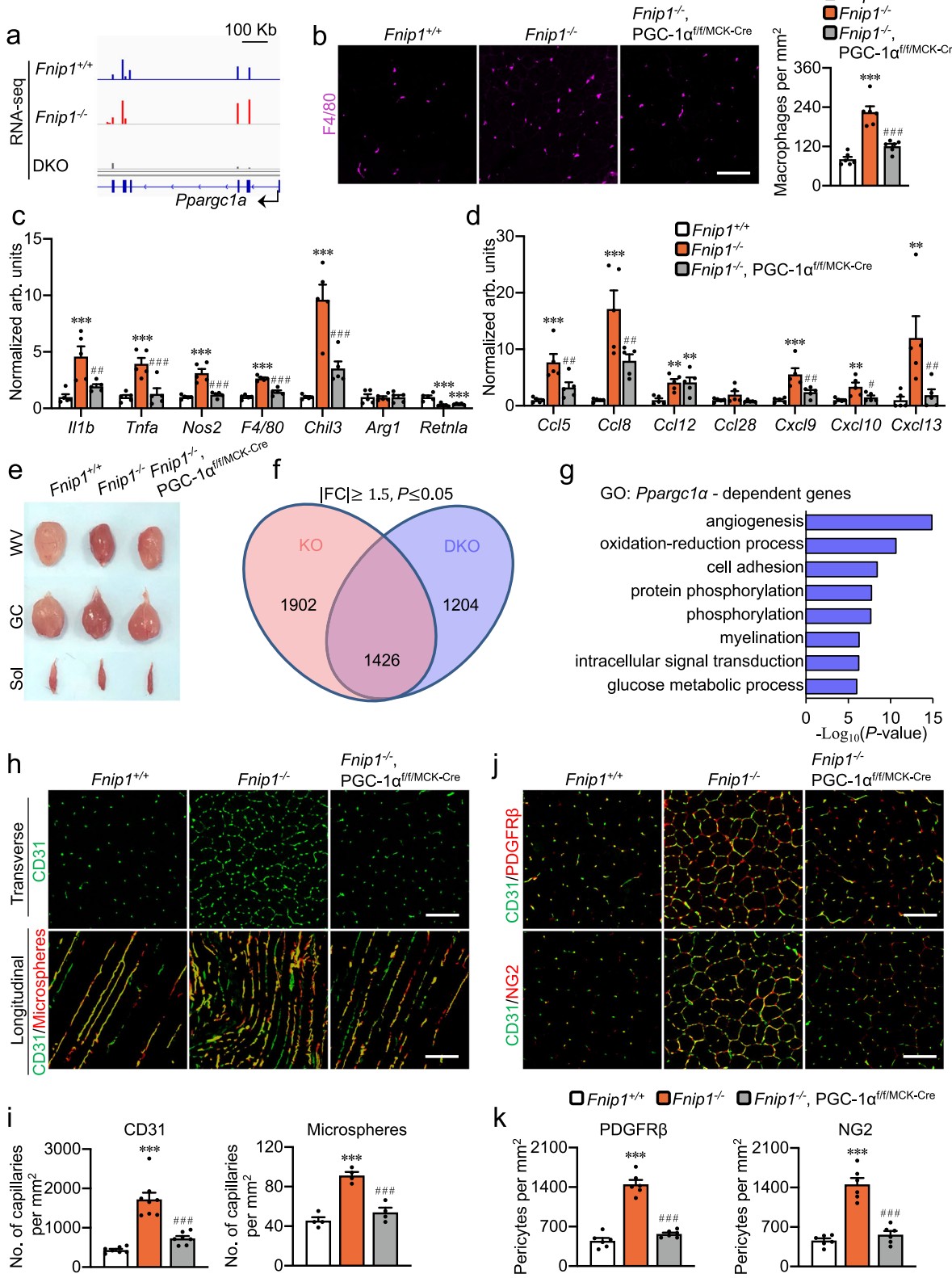

vessels compared to fast-glycolytic myofibers[51], we speculate that loss of myofiber FNIP1 promotes and coordinates vascular supply and metabolic demand in oxidative type I slow-twitch muscles. This elegantly links the regulation of consumption of oxygen by mitochondria to the delivery of oxygen and nutrients by the vasculature.

Angiogenesis is highly complex, involving multiple cell types and signals that must be coordinated in both space and time[7,8]. Although

previous study using whole-body knockout mice has provided a clue that FNIP1 is involved in skeletal muscle angiogenesis[22]. However, it is unclear whether the muscle capillary density changes are autonomous to endothelial cells or effects driven by deletion of *Fnip1* in myofibers. Our data indicate that myofiber FNIP1 coordinates the complex interactions between multiple cells types in skeletal muscle. On the basis of the data presented here, the recruited macrophages act, at least in part,

**Fig. 6 | FNIP1-dependent regulation of macrophage recruitment and muscle angiogenesis by PGC-1α. a** Genome browser tracks of RNA-seq data were visualized in IGV. **b** Representative images and quantification of F4/80 immunofluorescent staining in TA muscles from 10-week-old mice. Scale bar, 100 μm. $n = 6$ biologically independent mice. $P$ value: <0.0001. **c, d** Expression of genes (qRT-PCR) involved in macrophage activation (**c**) and chemokine signaling pathway (**d**) in GC muscles from 10-week-old mice. $n = 5$ biologically independent mice. **c** *$P$ value: 0.0006, 0.0004, <0.0001, <0.0001, <0.0001, 0.0001, 0.0004; #$P$ value: 0.0052, 0.0009, <0.0001, <0.0001, 0.0003. **d** *$P$ value: 0.0005, <0.0001, 0.0063, 0.006, 0.0005, 0.0043, 0.0058. #$P$ value: 0.0088, 0.0071, 0.0067, 0.0165, 0.0095. **e** Representative WV, GC and soleus muscles from 8-week-old mice. $n = 3$ biologically independent mice. **f** Identification of PGC-1α-dependent and -independent genes regulated by FNIP1 with the cutoff fold change greater than 1.5 and a cutoff significant level of $P < 0.05$.

**g** GO enrichment analysis ("Biological Process_Direct" term) of PGC-1α-dependent gene transcripts regulated by FNIP1. **h, i** Representative images (**h**) and quantification (**i**) of CD31 immunofluorescent (green) and microsphere perfused (red) images in GC muscles of 8-week-old mice. Scale bar, 100 μm. CD31, $Fnip1^{+/+}$, $n = 7$; $Fnip1^{-/-}$, $n = 8$; $Fnip1^{-/-}$,PGC-1α$^{f/f/MCK-Cre}$, $n = 7$ biologically independent mice; microsphere, $n = 4$ biologically independent mice. *$P$ value: <0.0001. #$P$ value: <0.0001.
**j, k** Representative confocal images (**j**) and quantification (**k**) of CD31 (green) and PDGFRβ (red), CD31 (green) and NG2 (red) co-staining in GC muscles from 8-week-old mice. Scale bar, 100 μm. $n = 6$ biologically independent mice. *$P$ value: <0.0001. #$P < 0.0001$. All data are shown as the mean ± SEM. **$P < 0.01$, ***$P < 0.001$ vs. corresponding $Fnip1^{+/+}$ controls, #$P < 0.05$, ##$P < 0.01$, ###$P < 0.001$ versus FNIP1 KO, determined by one-way ANOVA coupled to Fisher's least significant difference (LSD) post hoc test. Source data are provided as a Source Data file.

to help orchestrate the formation of new blood vessels that are patent and functional in FNIP1-depleted muscles. Surprisingly, our manipulation of FNIP1 in skeletal muscle induced robust angiogenesis within 7 days. Evidence is emerging that macrophages are crucial regulator of vascular niche and angiogenesis in multiple animal models[46,52–54]. There is evidence that macrophages can rapidly impact vascular niche activation and EC behavior. Specifically, in a 3D tissue-engineered human blood vessel networks in vitro, one-day exposure to macrophages has been shown to enhance angiogenesis and increase vessel formation[55]. In addition, macrophages have also been shown to physically interact with blood vessels and contribute to fusion of sprouting vessels[56,57]. It is tempting to speculate that such macrophage-mediated EC cell niche activation and angiogenesis are very active and rapid in skeletal muscle. It will be of interest to investigate the precise cascade of events affected by myofiber FNIP1. Our results suggest that FNIP1 deficiency may act through PGC-1α to induce CDH5 gene expression to improve endothelial barrier function. It would seem likely that myofiber FNIP1/PGC-1α signaling drives a concordant activation in muscle angiogenesis and endothelial barrier function to induce the formation of functional blood vessels. However, we only identify no massive leakage and would not detect more subtle changes in blood vessel permeability due to the limitations of our current methods. Whereas our results provide significant evidence that myofiber FNIP1 deficiency triggers muscle angiogenesis program through PGC-1α-dependent macrophage recruitment, we show that disruption of AMPKα1/α2 in $Fnip1^{KO}$ muscle resulted in significantly reduced mitochondrial function[26] but, surprisingly, have no effect on $Fnip1$ deficiency-mediated functional angiogenesis in multiple muscle types. Evidence has emerged that FNIP1 could act as co-chaperone that regulates the chaperone function of Hsp90[29,30,40,41]. However, our results suggest that Hsp90 may not be involved in the regulation of $Ppargc1a$ mRNA levels, and we did not find evidence to suggest that PGC-1α protein may be a client of Hsp90 based on public database (https://www.picard.ch/downloads/Hsp90interactors.pdf). However, we cannot exclude the possibility that other Hsp90 client proteins may be involved in FNIP1-mediated regulation of muscle angiogenesis.

In this study, we used a murine model of hindlimb vascular occlusion, which has been used preclinically to identify mechanisms that can accelerate or decelerate revascularization of ischemic muscle[49]. Our data indicate the accelerated formation of new functional vessels in the FNIP1 MKO ischemic skeletal muscle. The robust induction of muscle revascularization by myofiber FNIP1 deficiency, and its critical function in response to limb ischemia, strongly implicates myofiber FNIP1 as a powerful regulator of the angiogenic response to ischemia. Skeletal muscle capillary density is a tightly regulated process, it will be of interest to investigate this FNIP1-dependent angiogenesis mechanism in skeletal muscle in the setting of aging. Skeletal muscle is not the only tissue that requires adequate blood flow to maintain function. Heart, liver, bone, and the brain are also critically dependent on blood flow. It will also be interesting to test whether downregulation of the FNIP1 pathway also improves the vasculature and blood flow into those tissues

as well. Although a number of proangiogenic growth factors have been identified to be critically involved in controlling muscle angiogenesis remodeling, as exemplified by vascular endothelial growth factor (VEGF), the use of VEGF alone seems to lead to immature, leaky blood vessels[37,58]. One approach has been proposed to target the regulator that could coordinate multiple muscle angiogenesis signals appropriately. On the basis of our observations, myofiber FNIP1 is a powerful antiangiogenic regulator that could be targeted for therapeutic angiogenesis. FNIP1 deficiency induced a net proangiogenic signal that promotes muscle angiogenesis in mice. Interestingly, the mRNA expression levels of $Vegfa120$ were not upregulated in FNIP1 KO muscles. The reason for this difference is not clear. This could reflect the complex $VEGFA$-post-transcriptional regulatory effect. Indeed, regulatory mechanisms, including but not limited to alternative splicing, miRNA targeting and mRNA stability, and upstream open reading frame have been shown to affect $Vegfa121$ mRNA levels[59]. It is tempting to speculate that such mechanisms are active in muscle such that the steady-state $Vegfa120$ mRNA levels are not upregulated in FNIP1-deficient muscles. Our data suggest that the angiogenesis phenotype in FNIP1 MKO mice were not due to macrophage-derived VEGF. We speculate that one likely mechanism could involve the activation of macrophage-derived $Tnfa$ and $Nos2$ expression. We have not fully delineated this interesting, albeit complex, mechanism. Future studies will be necessary to delineate this macrophage-dependent angiogenesis mechanism in skeletal muscle.

In summary, our results have unmasked a crucial role of myofiber FNIP1 in regulating skeletal muscle functional angiogenesis through macrophage recruitment. Given that functional revascularization is vital to improve blood flow recovery and limit the ischemic damage of tissue, targeted modulation of myofiber FNIP1 may represent a potential avenue for therapeutic intervention in peripheral arterial diseases and cardiovascular diseases.

## Methods

### Animal studies
All animals performed in this study were approved by the Institutional Animal Care and Use Committee at the Model Animal Research Center (MARC) of Nanjing University (Approval No. GZJ09). The experiments in this manuscript are in compliance with relevant guidelines and ethical regulations. WT C57BL/6J mice purchased from GemPharmatech Co., Ltd (Jiangsu, China) were used for exercise experiments. $Fnip1^{f/f}$ mice were generated by the transgenic mouse facility at the Model Animal Research Center of Nanjing University as previously described[26]. Briefly, the Cas9 system targeting the exon 6 of the C57BL/6J mice $Fnip1$ gene was applied. The loxP sites were introduced upstream and downstream of exon 6 (92 bp) of $Fnip1$ gene by homologous recombination. To generate mice with a myofiber-specific disruption of the $Fnip1$ allele, $Fnip1^{f/f}$ mice were crossed with mice expressing Cre recombinase under the control of a human skeletal actin (HSA) promoter (Jackson Laboratory; stock no. 006139) to achieve myofiber-specific deletion of $Fnip1$ (FNIP1 MKO). Generation of FNIP1 KO mice have been described elsewhere[26]. To

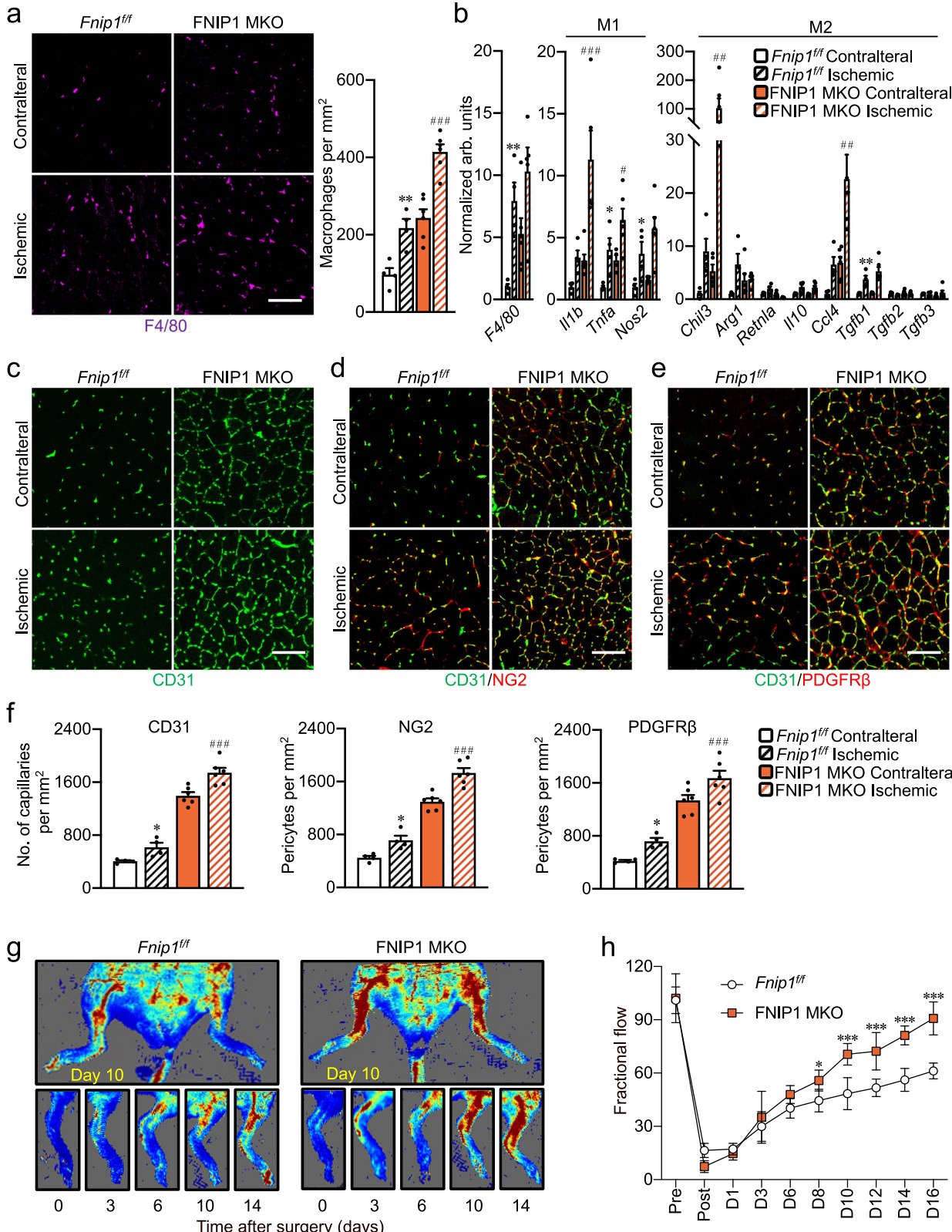

generate FNIP1 TgKO mice, male *Fnip1⁻/⁻* mice were first bred with female *Fnip1^{MCK-Tg+/−}* mice to generate *Fnip1^{+/−}*, *Fnip1^{MCK-Tg+/−}* males and *Fnip1^{+/−}* females, which were subsequently intercrossed to produce *Fnip1⁻/⁻*, *Fnip1^{MCK-Tg−/−}* (FNIP1 KO) and *Fnip1⁻/⁻*, *Fnip1^{MCK-Tg+/−}* (FNIP1 TgKO) and *Fnip1^{+/+}*, *Fnip1^{MCK-Tg−/−}* control littermates. To generate *Fnip1⁻/⁻*, AMPKα1/α2^{f/f/Myf5-Cre} (herein named TKO) mice, male *Fnip1⁻/⁻* mice were first bred with female *Ampka1/a2^{f/f Myf5-cre}* mice to generate

*Fnip1^{+/−}*, *Ampka1/a2^{f/+ Myf5-cre}* males and *Fnip1^{+/−}*, *Ampka1/a2^{f/+}* females, which were subsequently intercrossed to obtain male *Fnip1^{+/−}*, *Ampka1/a2^{f/f Myf5-cre}* and *Fnip1^{+/−};Ampka1/a2^{f/f}* females. Male *Fnip1^{+/−}*, *Ampka1/a2^{f/f Myf5-cre}* mice were finally bred with female *Fnip1^{+/−};Ampka1/a2^{f/f}* mice to produce *Fnip1⁻/⁻*, *Ampka1/a2^{f/f}* and *Fnip1⁻/⁻*, *Ampka1/a2^{f/f Myf5-cre}* and *Fnip1^{+/+}*, *Ampka1/a2^{f/f}* control littermates. To generate *Fnip1⁻/⁻*, PGC-1α^{f/f/MCK-Cre} (herein named DKO) mice, male

**Fig. 7 | Deletion of myofiber FNIP1 enhances hindlimb ischemia (HLI)-induced revascularization. a–f** Unilateral hindlimb ischemia was applied to both FNIP1 MKO and *Fnip1*^f/f littermate mice at the age of 14 weeks, and macrophage recruitment and neoangiogenesis were assessed on day 16 after the induction of hindlimb ischemia in both the contralateral and ischemic muscles. **a** Representative images of F4/80 immunofluorescent staining in GC muscles from indicated mice. Scale bar, 100 μm. Quantification of F4/80-positive macrophages per mm². *Fnip1*^f/f, *n* = 4; FNIP1 MKO, *n* = 6 biologically independent mice per group. *P value: 0.0027. #P value: <0.0001. **b** Expression of macrophage activation genes (qRT-PCR) in GC muscles from the indicated mice. *Fnip1*^f/f, *n* = 4; FNIP1 MKO, *n* = 6 biologically independent mice per group. *P value: 0.0074, 0.0195, 0.0252, 0.0092. #P value: 0.0009, 0.0379, 0.0089, 0.0017. **c–e** Representative confocal images of CD31 (green) and NG2 (red), CD31 (green) and PDGFRβ (red) co-staining in GC muscles

from the 14-week-old indicated mice. Fnip1f/f, *n* = 4; FNIP1 MKO, *n* = 6 biologically independent mice per group. **f** Quantification of capillaries and pericytes per mm². Fnip1f/f, *n* = 4; FNIP1 MKO, *n* = 6 biologically independent mice per group. *P value: 0.049, 0.046, 0.0162. #P value: <0.0001, <0.0001, <0.0001. **g** Representative laser Doppler images of 14-week-old *Fnip1*^f/f and FNIP1 MKO mice at the indicated times after hindlimb ischemia surgery. Fnip1f/f, *n* = 4; FNIP1 MKO, *n* = 5 biologically independent mice. **h** Quantification of blood flow in the ischemic limbs of *Fnip1*^f/f and FNIP1 MKO mice at the indicated times. Fnip1f/f, *n* = 4; FNIP1 MKO, *n* = 5 biologically independent mice. *P value: 0.0346, 0.0003, 0.0007, <0.0001, <0.0001. All data are shown as the mean ± SEM. *P < 0.05, **P < 0.01, ***P < 0.001 vs. contralateral, #P < 0.05, ##P < 0.01, ###P < 0.001 vs. corresponding *Fnip1*^f/f controls, determined by one-way (**a**, **b**, **f**) or two-way (**h**) ANOVA coupled to Fisher's least significant difference (LSD) post hoc test. Source data are provided as a Source Data file.

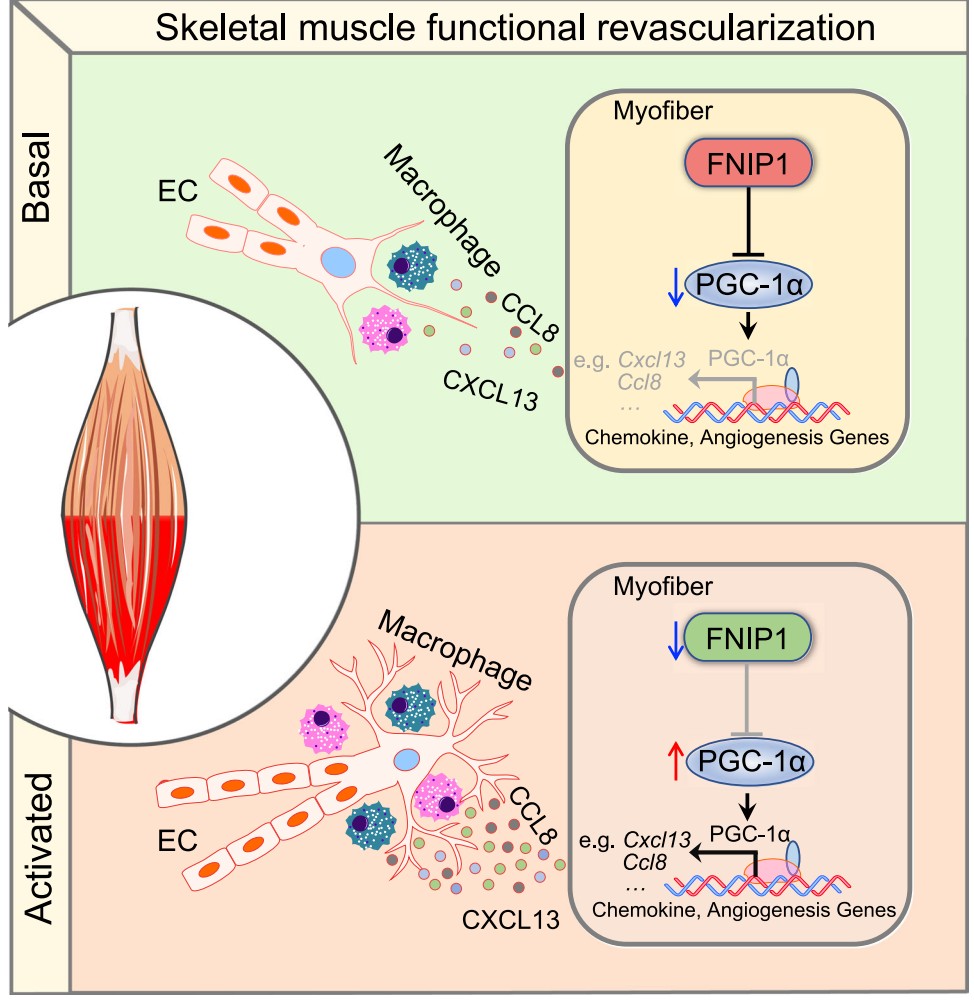

**Fig. 8 | Model of FNIP1-dependent regulation of skeletal muscle functional revascularization from ischemia.** The schematic depicts the proposed model for the orchestration of skeletal muscle functional revascularization by myofiber FNIP1.

*Fnip1*^−/− mice were first bred with female *Ppargc1a*^f/f MCK-cre mice to generate *Fnip1*^+/−, *Ppargc1a*^f/+ MCK-cre males and *Fnip1*^+/−, *Ppargc1a*^f/+ females, which were subsequently intercrossed to obtain male *Fnip1*^+/−, *Ppargc1a*^f/f MCK-cre and *Fnip1*^+/−, *Ppargc1a*^f/f females. Male *Fnip1*^+/−, *Ppargc1a*^f/fMCK-cre mice were finally bred with female *Fnip1*^+/−, *Ppargc1a*^f/f mice to produce *Fnip*^−/−, *Ppargc1a*^f/f and *Fnip1*^−/−, *Ppargc1a*^f/fMCK-cre, and *Fnip1*^+/+, *Ppargc1a*^f/f control littermates. To collect sera and tissue samples, mice were first anesthetized by isoflurane inhalation and then euthanized by cervical dislocation. Mice between 8 and 14 weeks of age (*n* = 3–11) were used for experiments. Mice were compared to their own littermates and the age of the mice are stated in the Figure legends. Mice were randomly assigned to various

analyses. Notably, both male and female mice were used in the study, muscle angiogenesis compared with littermate controls, were similar in female compared with male for FNIP1 KO lines. The animals were maintained with free access to pellet food (XieTong Biology, 1010082), and water in plastic cages at 21 ± 2 °C, relative humidity of 55 ± 10%, and kept on a 12 h light–dark cycle. All mice are harbored in the specific pathogen-free facility in Nanjing University.

**Hindlimb ischemia**

Hindlimb ischemia was performed in 8 to 14-week-old male *Fnip1*^f/f and FNIP1 MKO mice (4–6 biologically independent mice per group)[49,60]. Briefly, Animals were anesthetized and shaven anteriorly distal to the

midriff, including both hindlimbs. Unilateral mouse hindlimb ischemia was created by ligating the left femoral artery under general anesthesia (2–4% isoflurane in 100% oxygen at a flow rate of 1 L/min). A skin incision was made over the left femoral artery, from the knee towards the medial thigh. The femoral artery was then visualized and ligated proximally at the groin with double knots and again at the distal location close to the knee, and the segment of the artery between the two knots was removed. Cautery was used to control the bleeding. The incision was then closed with sutures and the mice were monitored throughout the recovery process. The recovery of blood flow at 1, 3, 6, 8, 10, 12, 14, and 16 days after surgery was tracked non-invasively by infrared Doppler scanning (Laser Doppler Perfusion Imager System, moorLDI-Mark 2, Wilmington, DE) of the lower limb. Perfusion was expressed as the ratio of the ischemic over the contralateral, non-operated leg. To minimize variables including ambient light and temperature and to maintain a constant body temperature, mice were exposed to infrared light for 10 min before laser Doppler scans.

### Histologic analyses
Eight to 14-week-old male and 8-week-old female mice muscle tissues were frozen in isopentane that had been cooled in liquid nitrogen. In all, 10 μm-thick serial muscles sections were cut in a Leica CM1850 cryostat at −20 °C and mounted on positively charged glass slides. Transverse sections collected from the widest part (mid-belly) of the gastrocnemius (GC), white vastus lateralis (WV) and tibialis anterior (TA) muscles were used for histological comparison to keep consistent between different mice. For IF stains, slides were fixed in ice-cold 4% PFA for 5 min, and permeabilized with ice-cold 0.5% Triton X-100-PBS for 10 min. These sections were then blocked using 5% normal goat serum (NGS)-PBS for 30 min at room temperature, followed by incubation with primary antibodies at 4 °C overnight. Three consecutive washes with PBS for 5 min each were followed by sequential incubation with appropriate secondary antibodies for 1 h at room temperature. Muscle sections were then washed three times with PBS for 5 min each. The staining images were captured under the confocal microscope. Vascular density and number of pericytes were quantified with ImageJ Fiji software (version 1.51, https://imagej.nih.gov/). For capillaries count in GC muscle, data were presented as the number of capillaries per $mm^2$ of the lateral part of GC to keep consistent between different mice.

### In vivo visualization of blood flow using fluorescent Microsphere
Blood vessel mapping was performed in 8 to 14-week-old male mice (*n* = 4–6 biologically independent mice per group)[61]. Briefly, to visualize in vivo blood flow, mice were anesthetized and perfused with 5–10 mL heparinized saline (10 U/mL) injected through the left ventricle. To identify capillaries with the capacity to be perfused through the circulatory system, a red fluorescent microsphere (0.1 μm, F8801, Invitrogen) suspension (1:20 dilution in heparinized saline) was intraventricularly perfused (10 mL, 1 mL/min) for 10 min, followed by euthanasia and tissue collection. Longitudinal cryosections (10 μm) of frozen gastrocnemius were processed and subjected to confocal microscopy for the visualization of perfused microvasculature.

### Evans Blue injections
Evans blue dye (1% solution Sigma-Aldrich) was i.p.-injected at a concentration of 1% volume per gram of body weight in 14-week-old male (*n* = 4 biologically independent mice per group). After 16 h, mice were sacrificed and skeletal muscle frozen histological sections were prepared. Evans blue incorporation was analyzed by fluorescence microscopy[62].

### AAV9 injection
AAV9s for in vivo expression of GFP or Cre were generated and provided by the Rongsen Gene Technology Co., Ltd (Jiangsu, China).

Briefly, recombinant AAV9s were generated to express the Cre recombinase or GFP under the control of the CAG promoter. AAV9s were subsequently generated using packaging plasmids pAAV-helper and pAAV9 together with pAAV-CAG-Cre or pAAV-CAG-GFP. AAV9s were diluted in 0.9% NaCl at $1 \times 10^{13}$ Vp/mL, and injected into muscles (30 μL per TA muscle) of 8-week-old mice. AAV9-GFP was used as a control, *n* = 9 biologically independent mice per group.

### Clodronate liposome treatment
Clodronate liposomes (YEASEN Biotechnology, Shanghai, China) were administered intraperitoneally 4 times (once every 2 days) at a dose of 200 μL per mouse. 8-week-old male mice (*n* = 5–8 biologically independent mice per group) were used. Before injection, remove Clodronate Liposomes and sterile PBS (for injection) from the refrigerator and allow them to warm to room temperature (18 °C) naturally, then invert 8–10 times to mix. Attach a 26-gauge needle to a 1 mL syringe and aspirate 200 μL of Clodronate Liposomes.

### TUNEL assay
Apoptosis of ECs was determined by TUNEL Bright Green Apoptosis Detection Kit (Vazyme, Nanjing, China) following the manufacturer's instructions. The images of TUNEL-positive signals were acquired with a laser scanning confocal microscopy (ZEISS LSM880), and quantified by ImageJ Fiji software (version 1.51, https://imagej.nih.gov/).

### Endothelial cell isolation and flow cytometry analysis
Endothelial cells from skeletal muscle were isolated from 10-week-old male FNIP1 MKO and Fnip1$^{f/f}$ control littermates (*n* = 4 biologically independent mice per group), MCK-FNIP1 Tg or NTG mice (*n* = 4 biologically independent mice per group)[10]. Briefly, hindlimb muscles were immediately dissected after euthanasia and minced into pieces on ice using a surgical blade. Then, the minced muscles were enzymatically digested in digesting buffer containing 2 mg/mL Collagenase IV (1886986, Gibco) and 2 mg/mL Dispase II (65558200, Roche) and 250 mM $CaCl_2$ in DPBS for 25 min at 37 °C with gentle shaking. Next, an equal volume of 20% FBS buffer in DPBS was added to stop the reaction. Thereafter, the suspension was passed through a series of 100-μm cell strainers (352350, Falcon) and 40-μm cell strainers (352340, Falcon) to remove tissue debris. After a series of centrifugation and washing steps, the cell pellets were resuspended in antibody solution with anti-mouse CD31 PE antibody (553373, BD Pharmingen, 1:200) and anti-mouse CD45 APC (103116, Bioligand, 1:200) antibody and incubated for 30 min at dark. After a series of washing steps with FACS wash buffer, the EC were sorted based on CD31$^+$CD45$^-$ staining by using FACS Aria III (BD Bioscience) sorter. Data were analyzed using FlowJo_v10.8.1 software (Tree Star) and the EC cell number was normalized to muscle mass.

### RNA in situ hybridization
The probe specific for *Fnip1* mRNA, used for in situ mRNA hybridization, was created by PCR amplification of cDNA from skeletal muscle using primers 5'-GGAAGACAGAGGAGTGGCTGAT-3' and 5'-TAA-TACGACTCACTATAGGGAGACCAACACCTCCTTTCCC-3'. Transcription of DIG-labeled antisense RNA probes was performed using standard methods. For acute bout of exercise, 8-week-old male fed WT C57BL/6J male mice (*n* = 3–4 biologically independent mice per group) were placed in an enclosed treadmill at a 10° incline and run for 10 min at 10 m/minute followed by a constant speed of 20 m/min for 90 min. RNA in situ hybridization was carried out as previously described[63]. Briefly, defrost muscle sections at room temperature and fixed in ice-cold 4% PFA for 10 min. Denature the probe and hybridization buffer (1:200 dilution) mix for 5–10 min at 70 °C. Then add denatured probe to sections and hybridize overnight at 65–70 °C. Prewarm wash solution (1xSSC, 50% formamide, 0.1% Tween-20) to 70 °C, wash 30 min for three times at 65–70 °C. Transfer slides to maleic acid buffer tween (MABT), wash 30 min for three times. Add 250 μL blocking solution to each slide

and incubate for 2 h at room temperature. Add anti-DIG AP antibody (11093274910, Roche, 1:5000 dilution) solution to the slides and incubate overnight at room temperature in dark. Transfer slides to Coplin jars and wash in MABT 4–5 times for 20 min at room temperature. Add NBT/BCIP (11681451001, Roche, 3.5 μL/mL) to the cool polyvinyl alcohol solution. Add slides and incubate at 37 °C overnight covered with foil. Wash for at least 6 h in PBST, changing solution every hour.

### RNA-Seq studies

Transcriptomics analyses were performed using RNA-sequencing[64,65]. Total RNA was isolated from the entire gastrocnemius muscle of 8-week-old male $Fnip1^{+/+}$, FNIP1 KO, FNIP1 TgKO, $Fnip1^{-/-}$, AMPKα1/α2$^{f/f/Myf5-Cre}$ (herein named TKO) and $Fnip1^{-/-}$, PGC-1α$^{f/f/MCK-Cre}$, FNIP1 MKO and $Fnip1^{f/f}$ control mice using RNAiso Plus (Takara Bio). RNA-seq using Illumina HiSeq 4000 was performed by Beijing Novogene Bioinformatics Technology Co., Ltd. Two independent samples per group were analyzed. Paired-end, 150-nt reads were obtained from the same sequencing lane. Transcriptome sequencing libraries averaged 39 million paired reads per sample, with 87.1% alignment to the mouse genome (UCSC mm10). The sequencing reads were then aligned to the UCSC mm10 genome assembly using TopHat 2.0.14 with the default parameters. Fragments Per Kb of exon per million mapped reads (FPKM) were calculated using Cufflinks 2.2.1. The criteria for a regulated gene were a fold change greater than 1.5 (either direction) and a significant $P$ value (<0.05) versus control. For pathway analysis, the filtered data sets were uploaded into DAVID Bioinformatics Resources 6.8 to review the bio pathways using the Functional Categories "Biological Process_ Direct" term, and the regulated terms were ranked by $P$ value. Notably, the FNIP1/PGC-1α regulation of muscle angiogenesis identified in the unbiased global RNA-seq analyses was validated by RT-PCR and functional assays. The RNA-seq data have been deposited in the National Genomics Data Center (NGDC) Genome Sequence Archive and are accessible through GSA Series accession number: CRA008213 and CRA008211. The heatmap analysis of regulated genes was generated by using R software (Version 3.3.2) and ggplot2/gplots package.

### RNA analyses

Quantitative RT-PCR was performed, briefly, total RNA was extracted from entire GC or TA muscle using RNAiso Plus (Takara Bio). The purified RNA samples were then reverse transcribed using the PrimeScript RT Reagent Kit with gDNA Eraser (Takara Bio). Real-time quantitative RT-PCR was performed using the ABI Prism Step-One system with Reagent Kit from Takara Bio. Specific oligonucleotide primers for target gene sequences are listed in Supplementary Table 1. Arbitrary units of target mRNA were corrected to the expression of $36b4$.

### Antibodies and immunoblotting studies

Antibodies directed against PE-anti-CD31 (553373, 1:200 dilution) and CD31 (550274, 1:200 dilution) were from BD Biosciences. Antibodies directed against mPDGFRβ (AF1042, 1:40 dilution) were from R&D Systems, antibodies directed against Brilliant Violet 421-anti-F4/80 (123132, 1:200 dilution), Alexa Fluor 647-anti-CD206 (141712, 1:200 dilution), FITC-anti-CD80 (104705, 1:200 dilution) and APC/Cyanine 7-anti-CD45 (103116, 1:200 dilution) were from Biolegend, antibodies directed against Ki67 (ab15580, 1:500 dilution) was from Abcam, antibodies directed against α-tubulin (bs1699, 1:5000 dilution) were from Bioworld; antibodies directed against NG2 (55027-1-AP, 1:200 dilution), CDH5 (27956-1-AP, 1:1000 dilution) and Hsp90 (13171-1-AP, 1:1000 dilution) were from Proteintech; antibodies directed against Myoglobin (sc-25607, 1:1000 dilution) were from Santa Cruz Biotechnology; antibodies directed against PGC-1α (1:1000 dilution) was developed in the laboratory of Daniel Kelly as described previously[66]; anti-FNIP1 was developed in the laboratory of Zhenji Gan with the help with Abcam (ab236547, 1:500 dilution). The secondary antibody Alexa Fluor 488 (A-11006, 1:400 dilution), Alexa Fluor 594 (A-21209, 1:400 dilution), Alexa Fluor 568 (A-11011, 1:400 dilution) was from Invitrogen; Alexa Fluor 488 (abs20026, 1:400 dilution) was from Absin. Western blotting studies were performed[67]. Blots were normalized to α-tubulin. The total protein concentration was measured by BCA assay using Pierce BCA Assay Kit Protocol (ThermoFischer Scientific). Equal total protein was loaded to each lane.

### Cell transfection and luciferase reporter assays

pSG5, pSG5-ERRβ, pcDNA3.1 and pcDNA3.1-PGC-1α vectors have been described previously[68,69]. The mouse $Cxcl13$ gene promoter was generated by PCR amplification from C57BL/6J genomic DNA followed by cloning into the pGL3 Basic luciferase reporter plasmid using BglII and HindIII sites. The following cloning primers were used: Fw 5'-CTAGCCCGGGCTCGAGATCTAAGATGGAGCGGGGCTTTGT-3', Rv 5'-AGTACCGGAATGCCAAGCTTGGAGTCTGTGCGTGCATTCC-3'. All constructs were confirmed by DNA sequencing. HEK293T (CRL-3216) and C2C12 cells (CRL-1772) were obtained from the American Type Culture Collection, and were cultured at 37 °C and 5% $CO_2$ in Dulbecco's modified Eagle's medium supplemented with 10% fetal calf serum, 1000 U/mL penicillin and 100 mg /mL streptomycin. For Hsp90 inhibition by 17DMAG treatment, after 3 days of differentiation, C2C12 cells were treated with Hsp90 inhibitor 17DMAG (0.1 μmol) for 24 h. For Hsp90 inhibition by siRNAs, the siRNAs specific to $Hsp90aa1$ were purchased from Thermo Fisher. siRNA transfection into C2C12 cells was performed using Lipofectamine 3000 Plus transfection reagent (Invitrogen) following the manufacturer's guidelines. Cells are harvested 48 h post transfection. Transient transfections in HEK293T cells were performed using PEI Transfection Reagent (Polysciences) following the manufacturer's protocol. Luciferase reporter assays were performed[69,70], briefly, 100 ng of reporter was co-transfected with 150 ng of pcDNA3.1-PGC-1α and 35 ng of pSG5-ERRβ and 25 ng of CMV promoter-driven Renilla luciferase to control for transfection efficiency. Cells were harvested 48 h after transfection. The luciferase assay was performed using Dual-Glo (Promega) according to the manufacturer's recommendations. All transfection data are presented as the mean ± standard error of the mean (SEM) for at least three separate transfection experiments.

### Statistical analyses

All studies were analyzed by Student's $t$ test or one-way ANOVA coupled to a Fisher's least significant difference (LSD) post hoc test when more than two groups were compared. No statistical methods were used to predetermine sample sizes, and sample size (range from $n = 3$ to $n = 11$) are explicitly stated in the figure legends. All data points were used in statistical analyses. Data represent the mean ± SEM, with a statistically significant difference defined as a value of $P < 0.05$.

### Reporting summary

Further information on research design is available in the Nature Portfolio Reporting Summary linked to this article.

## Data availability

The RNA-seq data reported in this paper have been deposited in the Genome Sequence Archive in National Genomics Data Center, China National Center for Bioinformation/Beijing Institute of Genomics, Chinese Academy of Sciences GSA: CRA008213 and CRA008211. Mouse genome (UCSC mm10) (http://www.genome.ucsc.edu). Source data are provided with this paper.

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

## Acknowledgements

This work was supported by grants from the Ministry of Science and Technology of China (National Key R&D Program of China 2018YFA0800700 and 2022YFA0806000) to Z.G., National Natural Science Foundation of China (No. 31922033, 91857105, 32071136, 32100922 and 32100942) to Z.G., T.F., and D.X., Fundamental Research Funds for the Central Universities (021414380517) to T.F., (021414380524, 021414380529 and 021414380511) to Z.G., Project funded by China Postdoctoral Science Foundation (2023M731633, 2021M691524 and 2022T150313) to Y.Y. and D.X., Natural Science Foundation of Jiangsu Province Grant (BK20230146) to Y.Y., Ministry of Science and Technology of China (National Key R&D Program of China 2022YFA1104300 and 2021YFA1101902) to S.H. This work was also supported by Collaborative Innovation Center of Food Safety and Quality Control in Jiangsu Province, Jiangnan University (2022-3-1).

## Author contributions

Z.S., L.Y., A.K., and J.Y. designed and performed most of the experiments. Z.S., L.Y., and T. F. analyzed the data, and wrote the manuscript. Z.Y., L.X., Y.Y., J.L., Y.M., D.Z., D.X., H. Y., Z. Z., Y.J., C.D., Q.G., and T.F. participated in the collection and analysis of the data. H. W., Y.L., L.W., and S.H. contributed reagents and provided scientific insight and discussion. Z.G., T.F., and S.H. conceptualized and interpreted the experiments. Z.G. wrote the manuscript and supervised the work. All authors reviewed and contributed to the manuscript.

## Competing interests

The authors declare no competing interests.
