## [Peer Review File · Nature Communications]

REVIEWER COMMENTS

Reviewer #1 (Remarks to the Author):

What appears to be the driving question of the manuscript is the relationship between myocytes (and in particular myocyte derived FNIP1 folliculin interacting protein 1 and the capillary density in skeletal muscle. Though not entirely, the models largely focus on myocyte derived knock-out and over-expression studies. I will provide suggestions to the authors in two ways. First I will comment from front to back (not necessarily prioritized) .

1) There are syntax errors even in the abstract.

2) The introduction, paragraph 1 has two "issues." The pathophysiology of brain vs. muscle ischemia is so vastly different and brain ischemia is not likely to be treatable by angiogenesis. Exactly what is functional angiogenesis?

3) Finding a higher capillary density that is expected or shown in the comparator groups is not by definition angiogenesis. Going to much of Figure 1. Is the data in Fig 1 particularly 1b and 1c due to differences in EC proliferation, or cell death, or both. Identification of the driving process would allow more detailed vs. a cursory investigation. The models are driven by the myocyte expression but what are the expression levels in other "key" cells?

4) Page 7 132 VEGFB ---often called large B which is a completely different gene from VEGFa or VEGF(isoform length)b -with b being within the a gene family but little b which can only be distinguished from a with specific probes

5) Page 8 line 159 – nice data suggestive of FNIP1 and vascular leakiness --- the mechanism should be explored to some extent, not just pathways

6) Page 8 to 9 -- Lots of genes are differentially expressed by bulk RNA-seq --- would it not be ideal to know how differences in EC number potentially contribute or even better isolate EC from the muscle to look at EC specific gene expression.

7) Figure 2 – H cannot be used for what it is stating to measure. note the differences in signal in the tail.

8) Did the authors look at myoglobin?

9) Figure 3D – potentially interesting data is being overlooked --- within part d "why" is VEGF121 not upregulated, -- VEGFb or VEGFB. What about angiogenesis inhibitors?

10) Lines 286-291 --- a single gene is sufficient or solely responsible.

Reviewer #2 (Remarks to the Author):

In this manuscript by Sun et al, the authors attempted to decipher the mechanisms of functional angiogenesis in skeletal muscle. Skeletal muscle revascularization from ischemia is a very complex process involving interactions between several cell types and signals within the muscle microenvironment, including myofibers, endothelial cells and macrophages. The authors have shown that the co-chaperone FNIP1 expression in muscles is down regulated in response to exercise training. Using both gain-of-function and loss-of function cell type-specific genetic models, they showed that FNIP1 is a negative regulator of angiogenesis in muscle. Myofiber specific down-regulation of FNIP1 enhanced the formation of patent, functional, nonleaky blood vessels. Very importantly, this process appears to be independent of AMPK. The authors have shown that Myofiber FNIP1 deficiency induces PGC-1alpha to activate chemokine genes transcription, thereby driving the macrophage recruitment and muscle angiogenesis program. Deletion of myofiber FNIP1 improved the recovery of blood flow in the murine hindlimb ischemia model of peripheral artery disease. The authors concluded that FNIP1-macrophage signaling axis to control functional angiogenesis in skeletal muscle. These findings are significant and provide additional functional information towards FNIP1. The authors provide convincing high-quality data with addition of appropriate statistical analysis to support their claims. This manuscript is worthy of publication in the Nature Communications. However, there are few comments that the authors need to address.

Figures- 1-4- conclusively demonstrates the role of FNIP1 in regulation of macrophage recruitment and muscle angiogenesis. The RNA-seq data are not accessible from the database until 2024-09-19. It is impossible to evaluate the raw data presented in Figures 1 & 2.

Figure 5- demonstrates FNIP1 mediated regulation of angiogenesis in muscles through PGC-1alpha- and eventual macrophage recruitment. However, the exact mechanism of FNIP1 mediated regulation of PGC-1alpha remains absent in both the Result and Discussion sections. The authors have not made any attempts at least to consider or speculate the involvement of hyperactive molecular chaperone Hsp90 in upregulation of PGC1alpha (even at the transcriptional level) or any other client proteins involved in angiogenesis.

Figure 6- male vs female mice been considered in this study?

Figure 8- Does AMPK play a role in this signaling?

Reviewer #3 (Remarks to the Author):

In this manuscript, the authors investigate the role of FNIP1 in skeletal muscle angiogenesis. They find the KO of FNIP1 has increased vascular density and this is rescued by PGC1alpha KO but not by AMPK KO. They then investigate the role of recruited macrophages in the process. It was already known that the FLCN KO has increased vascular density that is rescued by mating with PGC1alpha KO, but not by rapamycin (Hasumi et al, 2012). In that paper, angiogenic genes were analyzed and not vascular density, but the changes in the muscle color that were presented made it clear that there was more vascularization. The fact that the FNIP1 has a similar role to FLCN is novel, but also not that surprising. The real added insight from this current submission is that the mechanism is based on immune cell recruitment.

Though the role of immune cells downstream of FNIP1 is very interesting, there are significant problems with the experiments that must be rectified.

Major:

Figure 1a shows a reduction in mRNA and protein with exercise but this is done in whole tissue. Which cells are expressing FNIP1? Do all expressing cells downregulate FNIP1? I realize from the western blots that the available antibody is non-specific and cannot be used for immunostaining. This could however be evaluated by in situ hybridization or RNAScope. This would also help in interpreting the TgKO FNIP1 and the MKO mice phenotypes.

The age of mice is not consistently reported. The methods say "8-14 weeks", but then one experiment is suddenly with 4-week-old mice. The age should be reported in the legend for every figure.

The ALP staining adds little to manuscript and is not a traditional vascular marker. It should be removed, unless the authors can explain how it stains for something other than vessel density.

Quantification that are "per HFP" are not acceptable since it does not allow for comparison with other published work. All quantification needs to be converted to "per μm^2 " or "per mm^2 ".

Microsphere injection as performed is not a recognized method to measure vessel leakiness, especially considering the resolution of the current images and the lack of co-staining for an endothelial cell marker. Extravasation of microsphere from leaky vessels occurs within microns of the vessel. The traditional method is to inject microspheres through the tail vein and then flush the vasculature with either PBS or PGA to wash out the beads within 5-10 minutes. The tissue is then costained with endothelial cell marker. Beads that remain after flushing had extravasated. The current method would only identify massive leakage and would not detect more subtle changes in permeability. This is not sufficient to claim that there is no permeability issues with the vessels.

Similarly, Evans blue needs an endothelial cell co-stain to normalize the Evans blue to the vascular density.

There are not enough details for the microsphere method. How long were they allowed to circulate? I also believed the authors meant to write 0.1 μm not 0.1mM, but that is not clear.

Pericytes need two markers to be identified. PDGFRB also marks smooth muscle cells. Furthermore, the quantification should be normalized to vessel density. Therefore, the stainings need to be repeated with an endothelial cell marker and a second pericyte marker.

In the methods, the authors indicate that control mice are littermates. The methods also says that WT mice were purchased from GemPharmaTech. In the paper, control mice are listed as WT, which would indicate that they are not at all littermates (but instead purchased). The only way mice can be from the same litter is if the controls are heterozygotes (and not wildtype). The authors need to name the control/experimental by genotype. Hets should be indicated as “FNIP1+/-“ or just “FNIP1 Het” and not as WT. If the control are indeed not littermates but purchased mice, then the experiments need to be repeated. Skeletal muscle density varies too much by strain to simply purchased separate control animals.

In the same vein, it is somewhat difficult to figure out how all these mice can be obtained as littermates. Listing the genotypes of the parents that were used to produce the experimental mice in the methods (or supplemental methods). Example “Het, TgKO and KO mice were obtained by mating FNIP1 tg/+ FNIP +/- mice with FNIP -/- mice”.

In supplemental figure 1A, FNIP1 dependent and independent gene expression changes are identified. How can FNIP dependency be measured with the current experimental setup? I believe that this should be myocyte-dependent and independent (and therefore is just a typo), but I left this in “major” comments in case that is not the case.

The first paper from the authors on the TgKO had a different list of processes identified in the FNIP1-dependent GO analysis (Xiao et al, Plos Genetics, 2021). Angiogenesis did not show up then. The description in the legend for that paper and this one are identical. The original paper stated 12-week-old mice and this paper states 8 to 14 weeks. Can the authors explain why the GO analysis is completely different?

All the bulk RNA-Seq is performed with n=2 per group. The authors state that there are 39 million reads per sample. And while this can be analyzed, and would be acceptable if it was for the analysis of a difficult to obtain tissues (such as a human biopsy), there is very little justification for not doing three biological replicates when working with mouse tissues.

In the FNIP1KO AMPKa1a2 f/fMyh5Cre mice, the loss of AMPK normalizes VEGF164 levels but does not normalize vascular density. Only the 188 isoform and Vegfb remain high. These are relatively minor players in angiogenesis. Have the authors investigated factors from outside the VEGF family that could explain the increase in vascular density?

When analyzing the presence of macrophages by qPCR, the authors use only markers of M1 macrophages (iNOS, TNF α , IL1b, etc.). And yet the phenotype is one of angiogenesis. Given that the most novel part of this manuscript is that the effect of FNIP1 KO in myocytes is to affect macrophage recruitment, the relative abundance of different subtypes of macrophages should be analyzed (by FACS, immunostaining, in situ, single cell RNA-Seq.. there are many options). M2-like markers such as IL10, CCL4, Ym1, TGF β should be included.

The authors do not list which serotype is used in the methods of the text. I eventually found it in the experimental schematics shown in Figure 4c and 4f. The authors should just refer to it as AAV9 consistently. Furthermore, what was the promoter for construct? How old were the mice at the time of injection?

In the data reported in figure 4f, the amount of angiogenesis that has occurred within 7 days of injection is almost as high as what was observed after 8 weeks. I am wondering if my interpretation of the protocol as shown in 4f is correct. The mice receive AAV9 injection and chlodronate at D0, and then every 2 days afterwards, chlodronate liposomes are reinjected, until the animals are sacrificed at D7. If this is not correct, the figure 4f should be improved. Could the authors propose how a reasoning for such a huge increase within 1 week?

Injecting AAV9-CCL8 and AAV9-CXCL13 barely increase vascular density (when compared to the increase seen in the FNIP1-MKO). Furthermore, these results represent pure correlation. A increases vascular density, B increases vascular density.. therefore A must be using B to increase vascular density. This is faulty logic. Furthermore, these experiments were done with 4-week-old mice, which makes it a developmental model rather than an adult model. The rest of the work uses adult models. I would remove these experiments. They need to be redone with the correct stage of mouse, but even then, they don't really add much.

The authors use luciferase system to show that PGC1 α can induce CXCL13. Why not do a ChIP assay. This would be a direct assay and not depend on finding the correct co-factors.

The authors results imply that macrophage derived VEGF drives the phenotype. What are the VEGF levels after the chlodronate liposome injection. This rescue could be very informative for what factors are being secreted by the macrophages that induce angiogenesis.

Given that PGC1a KO rescues the phenotype, why are all the differentially regulated cytokines identified in 5C tested in the FNIP1KO PGC-1a f/fMCKCre mice? A different list of genes are tested. This could be very informative to identify which factor secreted by the myocytes is needed to recruit immune cells.

Minor

The authors use the term “master regulator” on many occasions. That is a loaded term and should really be avoided. It implies that that factor can override all other factors, which rarely (or never) exists in biology.

Spelling: agammaglobulinemic should be agammaglobulinemia.

On line 136, the authors refer to “fig 1g,f”. It should be “1g,h”.

What are the units of figure 2i?

The authors list the antibody for pericytes as “mPDGF Rb”. It should be mPDGFRb. But more importantly, the antibody is stated to be AF1402. That is a typo, it should be AF1042.

Itemized Responses to Reviewer #1

“Reviewer #1: What appears to be the driving question of the manuscript is the relationship between myocytes (and in particular myocyte derived FNIP1 folliculin interacting protein 1 and the capillary density in skeletal muscle. Though not entirely, the models largely focus on myocyte derived knock-out and over-expression studies. I will provide suggestions to the authors in two ways. First I will comment from front to back (not necessarily prioritized)”

We sincerely appreciate the Reviewer’s critical and constructive review of our manuscript. Our itemized responses are as follows:

“1) There are syntax errors even in the abstract.”

We thank the Reviewer for the careful review. We have corrected grammar mistakes throughout the manuscript.

“2) The introduction, paragraph 1 has two "issues." The pathophysiology of brain vs. muscle ischemia is so vastly different and brain ischemia is not likely to be treatable by angiogenesis. Exactly what is functional angiogenesis?”

We have removed the brain ischemia, thanks. For the “functional angiogenesis”, we intend to emphasize the formation of patent, nonleaky, and pericyte-covered blood vessels, we have explained the functional angiogenesis in the revised Introduction text (page 4, line 68-69).

“3) Finding a higher capillary density that is expected or shown in the comparator groups is not by definition angiogenesis. Going to much of Figure 1. Is the data in Fig 1 particularly 1b and 1c due to differences in EC proliferation, or cell death, or both. Identification of the driving process would allow more detailed vs. a cursory investigation. The models are driven by the myocyte expression but what are the expression levels in other "key" cells?”

We thank the Reviewer. We have performed Ki67 staining assays to determine the proliferation of endothelial cells (EC) in skeletal muscles from NTG and MCK-FNIP1 Tg mice. We found that myofiber FNIP1 overexpression had no significant impact upon EC proliferation (new Supplementary Fig.1c). Interestingly, TUNEL staining analysis showed that overexpression of FNIP1 in myofiber leads to increased EC apoptosis in skeletal muscles from MCK-FNIP1 Tg compared to NTG mice (new Supplementary Fig. 1d). This indicates that overexpression of FNIP1 in myofiber may affect the EC cell niches.

Notably, the muscle creatine kinase promoter (kind gift of E.N. Olson, University of Texas Southwestern) was used to generate myofiber-specific *Fnip1* transgenic mice (MCK-FNIP1 Tg mice). The MCK-*Fnip1* transgene transcript was efficiently expressed in skeletal muscle, and we observed no change in *Fnip1* mRNA levels in the ECs isolated from MCK-FNIP1 Tg mice compared to NTG mice (new Supplementary Fig. 1b).

The above new data have been added to the revised new Supplementary Fig. 1b-d (page 7, line 131-135; line 141-146; Methods, page 28, line 648; page 30,31, line 691-712).

“4) Page 7 132 VEGFB ---often called large B which is a completely different gene from VEGFa or VEGF(isoform length)b -with b being within the a gene family but little b which can only be distinguished from a with specific probes”

We thank the Reviewer. There are two isoforms of VEGF-B (VEGF-B167 and VEGF-B186) due to alternative splicing. We have designed RT-PCR primers to assess the levels of transcripts specific to each of these two VEGF-B isoforms. We found that both mRNA levels of *Vegfb167* and *Vegfb186* were significantly up-regulated in the muscles of the FNIP1 KO mice but reduced in FNIP1 TgKO mice (revised Fig. 1f). Moreover, FNIP1 deficiency-mediated induction of *Vegfb167* and *Vegfb186* mRNA expression were significantly reduced in the absence of PGC-1 α but not AMPK α 1/ α 2 (revised Fig. 3d, Supplementary Fig. 6d). These new data have been added to the revised Fig.1f, Fig. 3d, Supplementary Fig 3e, Fig. 6d, and new Supplementary Fig. 8a.

“5) Page 8 line 159 – nice data suggestive of FNIP1 and vascular leakiness --- the mechanism should be explored to some extent, not just pathways”

Our data suggest that blood vessels induced by FNIP1 abrogation did not leak macromolecules such as Evans blue (EB) dye. We are interested in this, and we have performed additional experiments to explore the potential underlying mechanism. Vascular leakiness is regulated by mechanisms that control endothelial barrier function, and previous studies have identified the main component of endothelial junctions, VE-cadherin, that are crucial for endothelial barrier function (Claesson-Welsh L et al.. *Trends Mol Med.* 2021;27(4):314-331, PMID: 33309601; Schulte D et al.. *EMBO J.* 2011;30(20):4157-4170, PMID: 21857650; Frye M et al., *J Exp Med.* 2015;212(13):2267-2287, PMID: 26642851; Duong CN et al., *Arterioscler Thromb Vasc Biol.* 2020;40(2):378-393, PMID: 31826650; Corada M et al., *Proc Natl Acad Sci U S A.* 1999;96(17):9815-9820, PMID: 10449777.). We have further analyzed our RNA-seq data and conducted gene expression validation studies. Whereas the expression of many VE-cadherin genes was not changed in FNIP1 KO muscles versus controls, we found that VE-cadherin genes *cdh5* and *cdh20* mRNA levels were significantly increased in FNIP1 KO muscles (new Supplementary Fig. 7d,e). These results were of interest because the CDH5 has been shown to be critical for the maintenance of endothelial barrier (Claesson-Welsh L et al.. *Trends Mol Med.* 2021;27(4):314-331, PMID: 33309601 Frye M et al., *J Exp Med.* 2015;212(13):2267-2287, PMID: 26642851; Duong CN et al., *Arterioscler Thromb Vasc Biol.* 2020;40(2):378-393, PMID: 31826650; Schulte D et al.. *EMBO J.* 2011;30(20):4157-4170, PMID: 21857650). Western blotting further confirmed that the increased expression of CDH5 protein in FNIP1 MKO muscles (new Supplementary Fig. 7f). Moreover, muscle-specific deletion of PGC-1 α abolished the induction of CDH5 protein expression in FNIP1 KO muscles (new Supplementary Fig. 7g). Together, these new results suggest that FNIP1 deficiency may act through PGC-1 α to induce CDH5 gene expression to improve endothelial barrier function. It would seem likely that myofiber FNIP1/PGC-1 α signaling drives a concordant activation in muscle angiogenesis and

endothelial barrier function to induce the formation of functional blood vessels. We wish to establish the necessary assay systems in our independent studies to address this interesting FNIP1/PGC-1 α /CDH5 mechanism. This information has been added as new Supplementary Fig. 7 (page 18, line 397-409) and addressed in the revised Discussion (page 23, line 526-532).

“6) Page 8 to 9 -- Lots of genes are differentially expressed by bulk RNA-seq --- would it not be ideal to know how differences in EC number potentially contribute or even better isolate EC from the muscle to look at EC specific gene expression.”

We followed the Reviewer's suggestion, we have quantified ECs (CD31⁺, CD45⁻) using a fluorescence-activated cell sorting (FACS)-based method (Zhang J et al., *Cell Metab.* 2020;31(6):1136-1153.e7, PMID: 32492393). We observed a significant increase in the number of ECs from muscles of FNIP1 MKO mice relative to their littermate controls (revised Supplementary Fig. 3g,h). This is in line with enhanced muscle angiogenesis in FNIP1 MKO mice, and supports a role of FNIP1 in myofiber may affect the EC cell niches. We have also conducted RT-PCR analysis in ECs isolated from muscles of FNIP1 MKO mice. We found that EC-specific marker gene expression (*Pecam1*, *Nos3*, *Kdr*, *Fli1*, *Ets2* and *Cd34*) were not changed in ECs isolated from FNIP1 MKO muscles (revised Supplementary Fig. 3i). These results suggest that an increase in EC number contribute to the observed changes in gene expression by bulk RNA-seq. This information has been added as revised Supplementary Fig. 3g-i (page 10-11, line 219-227; Methods, page 30-31, line 696-712).

“7) Figure 2 – H cannot be used for what it is stating to measure. note the differences in signal in the tail.”

We appreciate the Reviewer's point. We have replaced the original Figure 2H with new representative images and re-quantified the data (revised Fig. 2l,m). We confirmed a significantly higher blood flow to the skeletal muscles in FNIP1 MKO muscles compared to controls (WT, 555.4 \pm 41.0 PU vs. FNIP1 MKO, 1020.1 \pm 83.5 PU, $p < 0.001$; $n = 7-8$ mice per group).

“8) Did the authors look at myoglobin?”

We found that the induced expression of myoglobin protein paralleled the activation of angiogenesis in FNIP1 MKO muscles (revised Supplementary Fig. 3f). The new data have been added to the revised Supplementary Fig. 3f (page 10, line 215-217).

“9) Figure 3D – potentially interesting data is being overlooked --- within part d "why" is VEGF121 not upregulated, -- VEGFb or VEGFB. What about angiogenesis inhibitors? ”

We thank the Reviewer. We have added new data demonstrated that mRNA levels of *Vegfb167* and *Vegfb186* were induced in both FNIP1 KO and TKO muscles (revised Fig. 3d). Interestingly, the mRNA expression levels of *Vegfa120* were not upregulated in FNIP1 KO muscles. We are also interested in this

observation. The reason for this difference is not clear. This could reflect the complex *VEGFA*-post-transcriptional regulatory effect. Indeed, regulatory mechanisms, including but not limited to alternative splicing, miRNA targeting and mRNA stability, and upstream open reading frame have been shown to affect *VEGFA121* mRNA levels (Arcondéguy T, et al. *Nucleic Acids Res.* 2013;41(17):7997-8010, PMID: 23851566; Elias AP, Dias S. *Microenviron.* 2008;1(1):131-139. PMID: 19308691; Dowhan DH, et al. *Mol Cell.* 2005;17(3):429-439. PMID: 15694343; Nowak DG, et al. *J Cell Sci.* 2008;121(Pt 20):3487-3495. PMID: 18843117; Bastide A, et al. *Nucleic Acids Res.* 2008;36(7):2434-2445. PMID: 18304943). It is tempting to speculate that such mechanisms are active in skeletal muscle such that the steady-state *Vegfa120* mRNA levels are not upregulated in FNIP1 KO muscles.

As suggested, we have also conducted studies to examine the possibility that FNIP1 regulates factors known to inhibit angiogenesis. While the expression of many antiangiogenic genes (*Ppargc1b*, *Thbs1*, *Thbs2*, *Vash1*, and *Endo1*) was not different in the FNIP1 KO compared with the control muscles, *Pedf* and *Plasminogen* mRNA levels were decreased in FNIP1 KO muscles (revised Supplementary Fig. 4a). Interestingly, total AMPK α 1/ α 2 loss reversed the *Plasminogen*, but not *Pedf*, expression in FNIP1 KO muscles (revised Supplementary Fig. 4a). Together, FNIP1 deficiency induced a net proangiogenic signal that promote muscle angiogenesis in mice.

The above points have been added as revised Supplementary Fig. 4a (page 12, line 263-269) and addressed in the revised Discussion (page 24-25, line 562-570).

“10) Lines 286-291 --- a single gene is sufficient or solely responsible.”

We agree with the reviewer. Our data suggest that activation of either *Ccl8* or *Cxcl13* is capable of activating the infiltration of macrophages in skeletal muscles. However, the Reviewer#3 think that those data are not conclusive, and suggest to remove them from the manuscript.

Itemized Responses to Reviewer #2

“Reviewer #2: In this manuscript by Sun et al, the authors attempted to decipher the mechanisms of functional angiogenesis in skeletal muscle. Skeletal muscle revascularization from ischemia is a very complex process involving interactions between several cell types and signals within the muscle microenvironment, including myofibers, endothelial cells and macrophages. The authors have shown that the co-chaperone FNIP1 expression in muscles is down regulated in response to exercise training. Using both gain-of-function and loss-of-function cell type-specific genetic models, they showed that FNIP1 is a negative regulator of angiogenesis in muscle. Myofiber specific down-regulation of FNIP1 enhanced the formation of patent, functional, nonleaky blood vessels. Very importantly, this process appears to be independent of AMPK. The authors have shown that Myofiber FNIP1 deficiency induces PGC-1alpha to activate chemokine genes transcription, thereby driving the macrophage recruitment and muscle angiogenesis program. Deletion of myofiber FNIP1 improved the recovery of blood flow in the murine hindlimb ischemia model of peripheral artery disease. The authors concluded that FNIP1-macrophage signaling axis to control functional angiogenesis in skeletal muscle. These findings are significant and provide additional functional information towards FNIP1. The authors provide convincing high-quality data with addition of appropriate statistical analysis to support their claims. This manuscript is worthy of publication in the Nature Communications. However, there are few comments that the authors need to address.”

We sincerely appreciate the Reviewer’s critical and constructive review of our manuscript.

“These findings are significant and provide additional functional information towards FNIP1. The authors provide convincing high-quality data with addition of appropriate statistical analysis to support their claims. This manuscript is worthy of publication in the Nature Communications.”

Thank you for these positive comments.

1. “Figures- 1-4- conclusively demonstrates the role of FNIP1 in regulation of macrophage recruitment and muscle angiogenesis. The RNA-seq data are not accessible from the database until 2024-09-19. It is impossible to evaluate the raw data presented in Figures 1 & 2.

The RNA-seq data discussed in this manuscript have been deposited in the Genome Sequence Archive in National Genomics Data Center, China National Center for Bioinformation/Beijing Institute of Genomics, Chinese Academy of Sciences (GSA: CRA008213 and CRA008211). The Reviewer link was provided in the original manuscript. Unfortunately, these links get expired after 3 months during the manuscript review process. We have now regenerated the new Reviewer links to allow the review of our RNA-seq data: CRA008213 (<https://ngdc.cncb.ac.cn/gsa/s/24hGCSIN>) and CRA008211 (<https://ngdc.cncb.ac.cn/gsa/s/22DHuXJ1>). Thanks.

2. “Figure 5- demonstrates FNIP1 mediated regulation of angiogenesis in muscles through PGC-1alpha- and eventual macrophage recruitment. However, the exact mechanism of FNIP1

mediated regulation of PGC-1 α remains absent in both the Result and Discussion sections. The authors have not made any attempts at least to consider or speculate the involvement of hyperactive molecular chaperone Hsp90 in upregulation of PGC1 α (even at the transcriptional level) or any other client proteins involved in angiogenesis.”

We appreciate the Reviewer’s insightful points. The exact mechanism of FNIP1-mediated regulation of PGC-1 α in skeletal muscle remains unclear. It is surely a very important and intriguing question that warrants serious further characterizations. Previously published studies have revealed that FNIP1 could act as co-chaperone that regulates the chaperone function of Hsp90 (Woodford et al., *Nat Commun.* 2016;7:12037, PMID: 27353360; Sager et al., *Cell Rep.* 2019;26(5):1344-1356.e5, PMID: 30699359; Sager et al., *Trends Biochem Sci.* 2018;43(12):935-937, PMID: 30361061; Backe SJ et al., *Cell Rep.* 2022;40(2):111039, PMID: 35830801). We agree with the Reviewer in that it would be interesting to test whether the chaperone function of Hsp90 regulates PGC-1 α . We have conducted additional experiments to address this intriguing question. First, we found that Hsp90 protein levels were not altered in FNIP1 MKO muscle (new Supplementary Fig. 7a) and we did not find evidence to suggest that PGC-1 α protein may be a client of Hsp90 based on a public database (<https://www.picard.ch/downloads/Hsp90interactors.pdf>). To further determine whether the chaperone function of Hsp90 is involved in the regulation of PGC-1 α , we have inhibited Hsp90 (by 17DMAG treatment) in C2C12 myotubes. We think it is worth noting that PGC-1 α protein levels are hardly detectable in C2C12 myotubes, and we thus measured the PGC-1 α mRNA levels. We have found that Hsp90 inhibition had no effect on PGC-1 α mRNA levels (new Supplementary Fig. 7b). The effect of Hsp90 siRNAs were also assessed, siRNA-mediated knockdown of Hsp90 failed to affect PGC-1 α mRNA levels (new Supplementary Fig. 7c). Together, these results suggest that Hsp90 may not involve in the regulation of PGC-1 α mRNA levels. However, we cannot exclude the possibility that other Hsp90 client proteins are involved in FNIP1-mediated regulation of muscle angiogenesis. Future studies will be necessary to explore this interesting FNIP1/Hsp90 mechanism in muscle angiogenesis. The above points have been added as new Supplementary Fig. 7a-c (page 17-18, line 391-397; Methods, page 34, line 794-800) and addressed in the revised Discussion (page 24, line 538-544).

3. “Figure 6- male vs female mice been considered in this study?”

We have not found any major phenotypic differences in the muscle angiogenesis results with the male versus female mice. The data in original Figure 6 showed muscle angiogenesis in male WT, FNIP1 KO and *Fnip1*^{-/-}, PGC-1 α ^{f/f/MCK-Cre} (DKO) mice. We have now provided muscle angiogenesis data generated from female WT, FNIP1 KO and DKO mice as revised Supplementary Fig. 6h,i, showing PGC-1 α -dependent regulation of muscle angiogenesis in FNIP1 KO mice of both sexes. Thus, muscle angiogenesis compared with littermate controls, were similar in female compared with male for FNIP1 KO lines. This point has been clarified further in the Methods on page 27, line 616-617.

4. “Figure 8- Does AMPK play a role in this signaling?”

We appreciate the Reviewer's point. We concluded that FNIP1-regulated muscle angiogenesis does not require AMPK based on the muscle angiogenesis phenotypes of our total AMPK α 1/ α 2 loss genetic mouse lines. We have removed the AMPK in Figure 8 model to clarify the AMPK-independent regulation of muscle angiogenesis by FNIP1.

Itemized Responses to Reviewer #3

“Reviewer #3: In this manuscript, the authors investigate the role of FNIP1 in skeletal muscle angiogenesis. They find the KO of FNIP1 has increased vascular density and this is rescued by PCG1alpha KO but not by AMPK KO. They then investigate the role of recruited macrophages in the process. It was already known that the FLCN KO has increased vascular density that is rescued by mating with PCG1alpha KO, but not my rapamycin (Hasumi et al, 2012). In that paper, angiogenic genes were analyzed and not vascular density, but the changes in the muscle color that were presented made it clear that there was more vascularization. The fact that the FNIP1 has a similar role to FLCN is novel, but also not that surprising. The real added insight from this current submission is that the mechanism is based on immune cell recruitment.

Though the role of immune cells downstream of FNIP1 is very interesting, there are significant problems with the experiments that must be rectified.”

We sincerely appreciate the Reviewer’s critical and constructive review of our manuscript. Our itemized responses are as follows:

Major comments:

1. *“Figure 1a shows a reduction in mRNA and protein with exercise but this is done in whole tissue. Which cells are expressing FNIP1? Do all expressing cells downregulate FNIP1? I realize from the western blots that the available antibody is non-specific and cannot be used for immunostaining. This could however be evaluated by in situ hybridization or RNAScope. This would also help in interpreting the TgKO FNIP1 and the MKO mice phenotypes.”*

We appreciate the Reviewers’ critical points. Following the Reviewer’s suggestions, we have conducted *in situ* hybridization experiments to stain *Fnip1* mRNA within skeletal muscles. Our data suggest that *Fnip1* mRNA is expressed in both myofibers and capillaries (new Supplementary Fig. 1a), which is consistent with previous reports that *Fnip1* is ubiquitously expressed (Baba M et al.. *Proc Natl Acad Sci U S A.* 2006;103(42):15552-15557, PMID: 17028174). Interestingly, expression of the *Fnip1* mRNA was reduced in both myofibers and capillaries after an acute bout of exercise (new Supplementary Fig. 1a). Moreover, efficient deletion of *Fnip1* in skeletal muscle by KO was also verified by *in situ* hybridization (new Supplementary Fig. 1e), and the MCK-*Fnip1* transgene transcript was expressed in a myofiber-specific manner (new Supplementary Fig. 1e). These new data have been added to the new Supplementary Fig. 1a,e (page 7, line 126-130; page 8, line 151-153; Methods, page 31-32, line 714-733).

2, *“The age of mice is not consistently reported. The methods say “8-14 weeks”, but then one experiment is suddenly with 4-week-old mice. The age should be reported in the legend for every figure.”*

We thank the reviewer. We used a wider age range of mice at the beginning of the study. We have not found any major phenotypic differences in the results with the impact of age, we have now stated the age of the mice in all the Figure legends.

3, *“The ALP staining adds little to manuscript and is not a traditional vascular marker. It should be removed, unless the authors can explain how it stains for something other than vessel density.”*

Alkaline phosphatase (ALP) staining can be used as an alternative marker for tissue vasculature, though it's not commonly applied. We have removed the data of ALP staining as requested.

4, *“Quantification that are “per HFP” are not acceptable since it does not allow for comparison with other published work. All quantification needs to be converted to “per μm^2 ” or “per mm^2 ”.”*

As suggested, we have re-analyzed (plotted the data per mm^2) those data and updated all the relevant Figures.

5, *“Microsphere injection as performed is not a recognized method to measure vessel leakiness, especially considering the resolution of the current images and the lack of co-staining for an endothelial cell marker. Extravasation of microsphere from leaky vessels occurs within microns of the vessel. The traditional method is to inject microspheres through the tail vein and then flush the vasculature with either PBS or PGA to wash out the beads within 5-10 minutes. The tissue is then costained with endothelial cell marker. Beads that remain after flushing had extravasated. The current method would only identify massive leakage and would not detect more subtle changes in permeability. This is not sufficient to claim that there is no permeability issues with the vessel.”*

We thank for these complimentary and constructive comments. We apologize that the description was not clear in the manuscript. To clarify this point, we would like to respectfully remind that we performed microsphere injection experiment using fluorescent Microsphere (100 nm) mainly for the visualization of perfused microvasculature, but not for the measurement of vessel leakiness. We have performed CD31 co-staining in microsphere injection experiment (revised Fig. 1g, Fig. 2d, Fig. 3e, Fig. 6h and Supplementary Fig. 3j,k) as suggested. These data indicate that the blood vessels induced by FNIP1 deficiency are patent and capable of sustaining blood flow. For vessel leakiness, our Evans blue (EB) dye infiltration test suggests that blood vessels induced by FNIP1 abrogation did not leak macromolecules EB (revised Supplementary Fig. 2e, Fig. 3l). In addition, please also see our response to the Reviewer#1 point #5 above. We have added new results to suggest that FNIP1 deficiency may act through PGC-1 α to induce VE-cadherin 5 gene expression to improve endothelial barrier function (new Supplementary Fig. 7d-g). It would seem likely that myofiber FNIP1/PGC-1 α signaling drives a concordant activation in muscle angiogenesis and endothelial barrier function to induce the formation of functional blood vessels. However, the Reviewer has pointed out the caveats and limitations regarding our current methods. We agree with the reviewers, the current method would only identify massive leakage and would not detect

more subtle changes in permeability. We have changed the language in the revised text to more accurately reflect the important point you raised. This information has been added as revised Fig. 1g, Fig. 2d, Fig. 3e, Fig. 6h, Supplementary Fig. 3j,k and new Supplementary Fig. 7d-g (page 8-9, line 171-177; page 11, line 229-230; page 17, line 384-385; page 18, line 397-409) and addressed in the revised Discussion (page 23, line 526-532).

6, *“Similarly, Evans blue needs an endothelial cell co-stain to normalize the Evans blue to the vascular density.”*

As suggested, we have added CD31 co-staining in our Evans blue (EB) dye infiltration test (revised Supplementary Fig. 2e, Fig. 3l) as suggested. We confirmed that blood vessels induced by FNIP1 abrogation did not leak macromolecules EB into myofibers.

7, *“There are not enough details for the microsphere method. How long were they allowed to circulate? I also believed the authors meant to write 0.1 μ m not 0.1mM, but that is not clear.”*

The Reviewer is correct that it is 0.1 μ m not 0.1 mM. We apologize for this error, which has now been fixed (page 8, line 171; page 29, line 664). We followed the published protocol for the *in vivo* visualization of blood flow using fluorescent Microsphere (Johnson C et al. *Circ Res.* 2004;94(2):262-268, PMID: 14670843). We have provided more details in the Methods section, including perfusion time (page 29, line 660-669). Briefly, to visualize *in vivo* blood flow, mice were anesthetized and perfused with 5-10 mL heparinized saline (10 U/mL) injected through the left ventricle. To identify capillaries with the capacity to be perfused through the circulatory system, we infused (10 mL, 1 mL/min) fluorescent microsphere (0.1 μ m, F8801, Invitrogen) at a 1:20 dilution in heparinized saline for 10 min, followed by euthanasia and tissue collection. Longitudinal cryosections (10 μ m) of frozen gastrocnemius were processed and subjected to confocal microscopy for the visualization of perfused microvasculature.

8, *“Pericytes need two markers to be identified. PDGFR β also marks smooth muscle cells. Furthermore, the quantification should be normalized to vessel density. Therefore, the stainings need to be repeated with an endothelial cell marker and a second pericyte marker.”*

We have added the CD31 and PDGFR β co-staining in skeletal muscles of FNIP1 MKO mice (revised Fig. 2f) as suggested. There was a positive correlation between capillary (CD31) and pericyte (PDGFR β) densities and pericyte coverage of endothelial cells was unchanged (revised Fig. 2f-h). Muscle sections were also stained for chondroitin sulfate proteoglycan 4 (CSPG4; also named as NG2), a second pericyte marker, and similar results were obtained when we conducted CD31 and NG2 co-staining (revised Fig. 2i-k). This supports that blood vessels induced by myofiber FNIP1 deletion were covered by pericytes. We have also added the CD31/ PDGFR β and CD31/NG2 co-staining in skeletal muscles of FNIP1 TgKO, DKO and TKO mice (revised Fig. 1i,j; Fig. 3f,g; Fig. 6j,k; Fig. 7d,e). These new data have been added as revised Fig. 1i,j; Fig. 2f-k; Fig. 3f,g; Fig. 6j,k; Fig. 7d,e (page 9, line 188-190; page 11, line 233-237; page 13, line 273-275; page 17, line 386-387; page 19-20, line 445-446).

9, “In the methods, the authors indicate that control mice are littermates. The methods also says that WT mice were purchased from GemPharmaTech. In the paper, control mice are listed as WT, which would indicate that they are not at all littermates (but instead purchased). The only way mice can be from the same litter is if the controls are heterozygotes (and not wildtype). The authors need to name the control/experimental by genotype. Hets should be indicated as “FNIP1+/-“ or just “FNIP1 Het” and not as WT. If the control are indeed not littermates but purchased mice, then the experiments need to be repeated. Skeletal muscle density varies too much by strain to simply purchased separate control animals.”

We think the Reviewer has raised an important point and we did not present our data clearly. The WT C57BL/6J mice purchased from GemPharmatech Co., Ltd were used for those exercise experiments. For all the genetically modified mice models used in this study, we conducted multiple rounds of mice breeding to make sure that the mice were compared to their own littermates to control for the relative differences in genetic background. We have provided more details with regard to mice breeding strategy in the Methods (page 26-27, line 587-588; 597-614). Specifically, to generate FNIP1 TgKO mice, male *Fnip1*^{-/-} mice were first bred with female *Fnip1*^{MCK-Tg+/-} mice to generate *Fnip1*^{+/-}, *Fnip1*^{MCK-Tg+/-} males and *Fnip1*^{+/-} females, which were subsequently intercrossed to produce *Fnip1*^{-/-}, *Fnip1*^{MCK-Tg-/-} (FNIP1 KO) and *Fnip1*^{-/-}, *Fnip1*^{MCK-Tg+/-} (FNIP1 TgKO) and *Fnip1*^{+/-}, *Fnip1*^{MCK-Tg-/-} control littermates. To generate *Fnip1*^{-/-}, AMPKα1/α2^{fl/flMyf5-Cre} (herein named TKO) mice, male *Fnip1*^{-/-} mice were first bred with female *Ampka1/α2*^{fl/flMyf5-cre} mice to generate *Fnip1*^{+/-}, *Ampka1/α2*^{fl/flMyf5-cre} males and *Fnip1*^{+/-}, *Ampka1/α2*^{fl/fl} females, which were subsequently intercrossed to obtain male *Fnip1*^{+/-}, *Ampka1/α2*^{fl/flMyf5-cre} and *Fnip1*^{+/-}; *Ampka1/α2*^{fl/fl} females. Male *Fnip1*^{+/-}, *Ampka1/α2*^{fl/flMyf5-cre} mice were finally bred with female *Fnip1*^{+/-}; *Ampka1/α2*^{fl/fl} mice to produce *Fnip1*^{-/-}, *Ampka1/α2*^{fl/fl} and *Fnip1*^{-/-}, *Ampka1/α2*^{fl/flMyf5-cre} and *Fnip1*^{+/-}, *Ampka1/α2*^{fl/fl} control littermates. To generate *Fnip1*^{-/-}, PGC-1α^{fl/flMCK-Cre} (herein named DKO) mice, male *Fnip1*^{-/-} mice were first bred with female *Ppargc1α*^{fl/flMCK-cre} mice to generate *Fnip1*^{+/-}, *Ppargc1α*^{fl/flMCK-cre} males and *Fnip1*^{+/-}, *Ppargc1α*^{fl/fl} females, which were subsequently intercrossed to obtain male *Fnip1*^{+/-}, *Ppargc1α*^{fl/flMCK-cre} and *Fnip1*^{+/-}, *Ppargc1α*^{fl/fl} females. Male *Fnip1*^{+/-}, *Ppargc1α*^{fl/flMCK-cre} mice were finally bred with female *Fnip1*^{+/-}, *Ppargc1α*^{fl/fl} mice to produce *Fnip1*^{-/-}, *Ppargc1α*^{fl/fl} and *Fnip1*^{-/-}, *Ppargc1α*^{fl/flMCK-cre}, and *Fnip1*^{+/-}, *Ppargc1α*^{fl/fl} control littermates. We have also relabeled the genotypes of the WT mice in the Figures to make this point clearer.

10, “In the same vein, it is somewhat difficult to figure out how all these mice can be obtained as littermates. Listing the genotypes of the parents that were used to produce the experimental mice in the methods (or supplemental methods). Example “Het, TgKO and KO mice were obtained by mating FNIP1 tg/+ FNIP +/- mice with FNIP -/- mice”.”

We thank the Reviewer and this was addressed in the point #9 above.

11, “In supplemental figure 1A, FNIP1 dependent and independent gene expression changes are identified. How can FNIP dependency be measured with the current experimental setup? I believe that this should be myocyte-dependent and independent (and therefore is just a typo), but I left this in

“major” comments in case that is not the case.”

Because *Fnip1* was only expressed in myocyte but not in other non-myocyte in FNIP1 TgKO muscles, we agree with the reviewer and we have changed “*FNIP1-dependent and independent*” to “*myocyte-dependent and independent*” to be more accurate. Thanks.

12, “The first paper from the authors on the TgKO had a different list of processes identified in the FNIP1-dependent GO analysis (Xiao et al, Plos Genetics, 2021). Angiogenesis did not show up then. The description in the legend for that paper and this one are identical. The original paper stated 12-week-old mice and this paper states 8 to 14 weeks. Can the authors explain why the GO analysis is completely different?”

We think the Reviewer has raised an important point and we did not present our data clearly. To be clear, these are the same mouse models as in Xiao et al, Plos Genetics, 2021. Total RNA isolated from the entire gastrocnemius muscle of 8-week-old male mice was used for RNA-seq. For the GO analysis in Xiao et al, *PLoS Genet.* 2021;17(3):e1009488, PMID: 33780446, the regulated genes were uploaded into DAVID Bioinformatics Resources 6.8. The Functional Annotation analysis using the “cellular compartments” defined by Gene_Ontology, Angiogenesis did not show up because “Angiogenesis” term is not belonged to “cellular compartments” that is not revealed in the analysis. In our current study, the regulated pathways were reviewed using the “Biological Process_Direct” term defined by Gene_Ontology. This analysis revealed that the primary regulated genes belong to the angiogenesis biological process (Fig. 1e). Indeed, we also found that several terms such as “oxidation-reduction process” and “lipid metabolic process” are significantly enriched in FNIP1-regulated gene dataset (Fig. 1e), which is consistent with our previous reports related to “cellular compartments” analysis (Xiao et al, *PLoS Genet.* 2021;17(3):e1009488, PMID: 33780446). We think that our results have uncovered a previously unrecognized role of FNIP1 that coordinately regulates mitochondrial function muscle fiber type and muscle angiogenesis. We have modified the descriptions in the revised Figure legends of Fig. 1e, Fig. 2c, Fig. 3c, and Fig. 6g to clarify this point.

13, “All the bulk RNA-Seq is performed with n=2 per group. The authors state that there are 39 million reads per sample. And while this can be analyzed, and would be acceptable if it was for the analysis of a difficult to obtain tissues (such as a human biopsy), there is very little justification for not doing three biological replicates when working with mouse tissues.”

We agree with the reviewer that an increase in sample size will help in this study. Unfortunately, because of the grant budget at the beginning of the study (2017) and the many mice groups were involved in this study, we used two independent muscle samples from each group for RNA-Seq. We would like to respectfully remind that FNIP1/PGC-1 α regulation of muscle angiogenesis, the findings resulted from the unbiased global RNA-seq analyses, was validated by the RT-PCR. In addition, the functional relevance of this FNIP1/PGC-1 α regulation of muscle angiogenesis has also been validated. We have also added a sentence to the revised Methods to clarify this point (page 32, line 751-753).

14, “In the *FNIP1KO AMPK α 1 α 2 f/fMyh5Cre* mice, the loss of AMPK normalizes VEGF164 levels but does not normalize vascular density. Only the 188 isoform and *Vegfb* remain high. These are relatively minor players in angiogenesis. Have the authors investigated factors from outside the VEGF family that could explain the increase in vascular density?”

We thank the Reviewer’s insightful point. We agree with the reviewer, as we have also added new gene expression data demonstrating that the expression of the *Vegf* genes was induced by FNIP1 deficiency but not affected by the clodronate treatment (see Figure 1a below). As further supported by the results of our new results, we speculate that factors from outside the VEGF family could be involved in the activation of angiogenesis by FNIP1 deficiency. Indeed, we have conducted further comparative analysis and found that the expression of many angiogenesis factors was induced by FNIP1 deficiency but not affected by the total loss of AMPK α 1/ α 2 in muscles (see Figure 1b below). Particularly, the *Tnfa* and *Nos2*, which are induced in FNIP1 KO muscles, have been shown to regulate vascular niche and angiogenesis (Sainson RC et al. *Blood*. 2008;111(10):4997-5007, PMID: 18337563; Hongu T et al. *Nat Cancer*. 2022;3(4):486-504, PMID: 35469015; Vågesjö E et al., *Circ Res*. 2021;128(11):1694-1707. PMID: 33878889; De Palma M et al. *Nat Rev Cancer*. 2017;17(8):457-474, PMID: 28706266). RT-qPCR confirmed that both *Tnfa* and *Nos2* gene expression were significantly induced in FNIP1 KO muscles but not affected by the total loss of AMPK α 1/ α 2 (see Figure 1c below). However, gene expression validation studies revealed that FNIP1 deficiency-mediated increased *Tnfa* and *Nos2* expression was reduced after clodronate treatment (see Figure 1d below). Moreover, we also confirmed the induction of *Tnfa* and *Nos2* expression was abolished in FNIP1 TgKO muscles (see Figure 1e below, revised Supplementary Fig. 5b), and muscle-specific disruption of PGC-1 α resulted in marked decrease in *Tnfa* and *Nos2* expression in the muscles of FNIP1 KO mice (see Figure 1f below, revised Fig. 6c). Together, the strong correlation between *Tnfa* and *Nos2* expression and muscle angiogenesis in our serious mouse models suggest that *Tnfa* and *Nos2* may be involved in FNIP1-regulated muscle angiogenesis. We have not delineated this interesting, albeit complex, mechanism. It will be of interest to conduct future studies to more precisely define this macrophage-dependent angiogenesis mechanism in skeletal muscle. This information has been added as new Supplementary Fig. 8 (page 18-19, line 409-426) and addressed in the revised Discussion (page 25, line 570-575).

Figure 1. Factors from outside the VEGF family could be involved in the activation of angiogenesis by FNIP1 deficiency. (a) Expression of genes (qRT-PCR) associated with angiogenesis in GC muscles from 8-week-old *Fnip1*^{fl/fl} mice injected with AAV9-Cre or control viruses following clodronate treatment. n = 4-7 mice per group. (b) Heatmap analysis of angiogenesis gene expression in skeletal muscle from indicated mice. (c, e, f) Expression of *Tnfa* and *Nos2* (qRT-PCR) in GC muscles from the indicated mice. n = 4-5 mice per group. (d) Expression of *Tnfa* and *Nos2* (qRT-PCR) in GC muscles from *Fnip1*^{fl/fl} mice injected with AAV9-Cre or control viruses following clodronate treatment. n = 4-7 mice per group. All data are shown as the mean ± SEM. **P* < 0.05 vs. corresponding controls, #*P* < 0.05 versus AAV9-Cre, determined by one-way ANOVA coupled to Fisher's least significant difference (LSD) post-hoc test.

15, "When analyzing the presence of macrophages by qPCR, the authors use only markers of M1 macrophages (iNOS, TNFa, IL1b, etc.). And yet the phenotype is one of angiogenesis. Given that the most novel part of this manuscript is that the effect of FNIP1 KO in myocytes is to affect macrophage recruitment, the relative abundance of different subtypes of macrophages should be analyzed (by FACS, immunostaining, in situ, single cell RNA-Seq.. there are many options). M2-like markers such as IL10, CCL4, Ym1, TGFβ should be included."

As suggested, we have conducted additional immunostaining to examine the relative abundance of different subtypes of macrophages. As predicted, the FNIP1 MKO muscle cryosections stained for the M1 macrophage marker CD80 or M2 macrophage marker CD206 showed an increase in both M1 and M2 macrophage recruitment compared with the control muscles (revised Fig. 4c,d). We have also included the results of RT-qPCR analysis of more M2-like markers genes (*Il10*, *Ccl4*, *Ym1/Chil3*, *Tgfb1-3*) from FNIP1 MKO muscle samples. Consistent with the immunostaining results, induction of both M1 and M2 macrophages marker genes expression was observed in muscle from FNIP1 MKO mice (revised Fig. 4b). Ischemia led to greater induction of both M1 and M2 macrophages marker genes expression in muscles from FNIP1 MKO mice (revised Fig. 7b). These results were added as revised Fig. 4b-d (page 13, line 290-293) and Fig. 7b (page 19 line 443-444).

16, “The authors do not list which serotype is used in the methods of the text. I eventually found it in the experimental schematics shown in Figure 4c and 4f. The authors should just refer to it as AAV9 consistently. Furthermore, what was the promoter for construct? How old were the mice at the time of injection?”

We apologize that the description of this part was not clear in the previous versions. We have provided more details with regard to AAV9 injection in the Methods section (page 29, line 676-682). In brief, we generated recombinant AAV9 to express the Cre recombinase or GFP under the control of the CAG promoter. AAV9s were subsequently generated using packaging plasmids pAAV-helper and pAAV9 together with pAAV-CAG-Cre or pAAV-CAG-GFP. AAV9s were diluted in 0.9% NaCl at 1×10^{13} Vp/mL, and injected into muscles (30 μ L per TA muscle) of 8-week-old mice. AAV9-GFP was used as a control.

17, “In the data reported in figure 4f, the amount of angiogenesis that has occurred within 7 days of injection is almost as high as what was observed after 8 weeks. I am wondering if my interpretation of the protocol as shown in 4f is correct. The mice receive AAV9 injection and clodronate at D0, and then every 2 days afterwards, clodronate liposomes are reinjected, until the animals are sacrificed at D7. If this is not correct, the figure 4f should be improved. Could the authors propose how a reasoning for such a huge increase within 1 week?”

The Reviewer is correct. Schematic illustrating the clodronate treatment in the presence of AAV9-Cre-mediated FNIP1 ablation in skeletal muscle in original Figure 4F. The mice receive AAV9 injection and clodronate treatment at D0, and then clodronate liposomes are reinjected once every two days until the animals are sacrificed at D7. Surprisingly, our manipulation of FNIP1 in skeletal muscle induced robust angiogenesis within 7 days. We are also very interested in this observation. Evidence is emerging that macrophages are crucial regulators of vascular niche and angiogenesis in multiple animal models (Willenborg S et al. *Blood*. 2012;120(3):613-625, PMID: 22577176; Hirose N et al. *Cell Transplant*. 2008;17(1-2):211-222, PMID: 18468252; Juhas M et al. *Nat Biomed Eng*. 2018;2(12):942-954, PMID: 30581652; Hongu T et al. *Nat Cancer*. 2022;3(4):486-504, PMID: 35469015). It is worth noting that there is evidence that macrophages can rapidly impact vascular niche

activation and EC behavior. Specifically, in a 3D tissue-engineered human blood vessel networks *in vitro*, one-day exposure to macrophages has been shown to enhance angiogenesis and increase vessel formation (Graney PL et al. *Sci Adv.* 2020;6(18):eaay6391, PMID: 32494664). In addition, macrophages have also been shown to physically interact with blood vessels and contribute to the fusion of sprouting vessels (Fantin A et al. *Blood.* 2010;116(5):829-840, PMID: 20404134; Hsu CW et al. *PLoS One.* 2015;10(7):e0131643, PMID: 26132702). We are interested in this, and we have performed additional experiments to explore the potential macrophage-mediated EC cell niche activation. Ki67 staining showed that AAV9-based FNIP1 ablation in *Fnip1^{fl/fl}* muscles leads to a dramatic induction of EC proliferation, and clodronate treatment resulted in marked reduction of EC proliferation in the muscles from *Fnip1^{fl/fl}* mice injected with AAV9-Cre (revised Supplemental Fig. 5h,i). Moreover, TUNEL staining analysis showed that FNIP1 deficiency-mediated decreased EC apoptosis was prevented in the presence of clodronate (revised Supplemental Fig. 5j,k). It is tempting to speculate that such macrophage-mediated EC cell niche activation and angiogenesis are very active and rapid in skeletal muscle. We believe that studies aimed at defining this macrophage-dependent muscle angiogenesis mechanism would be an important focus of study for the future. This information has been added as revised Supplementary Fig. 5h-k (page 15, line 322-329; Methods, page 28, line 648; page 30, line 691-694) and addressed in the revised Discussion (page 23, line 516-525).

18, “Injecting AAV9-CCL8 and AAV9-CXCL13 barely increase vascular density (when compared to the increase seen in the FNIP1-MKO). Furthermore, these results represent pure correlation. A increases vascular density, B increases vascular density.. therefore A must be using B to increase vascular density. This is faulty logic. Furthermore, these experiments were done with 4-week-old mice, which makes it a developmental model rather than an adult model. The rest of the work uses adult models. I would remove these experiments. They need to be redone with the correct stage of mouse, but even then, they don’t really add much.”

We have removed those AAV9-CCL8 and AAV9-CXCL13 experiments as suggested. Thanks.

19, “The authors use luciferase system to show that PGC1alpha can induce CXCL13. Why not do a ChIP assay. This would be a direct assay and not depend on finding the correct co-factors.”

Thank you for this query. Unfortunately, we were unable to assess PGC-1 α chromatin immunoprecipitation (ChIP) status due to the relative lack of sensitivity of the PGC-1 α antibody in house, which work for heart and FNIP1 MKO muscle western blot.

20, “The authors results imply that macrophage derived VEGF drives the phenotype. What are the VEGF levels after the clodronate liposome injection. This rescue could be very informative for what factors are being secreted by the macrophages that induce angiogenesis.”

We thank the reviewer. We do not believe that the angiogenesis phenotype in FNIP1 MKO mice was due

to macrophage-derived VEGF. We have added new gene expression data demonstrating that the expression of the *Vegf* genes was induced by FNIP1 deficiency but not affected by the chlodronate treatment (new Supplementary Fig. 8a). Please also see our response to the point #14 above. As supported by the results of our new results, we speculate that factors from outside the VEGF family could be involved in the activation of angiogenesis by FNIP1 deficiency. We have not fully delineated this interesting, albeit complex, mechanism. It will be of interest to conduct future studies to more precisely define this macrophage-dependent angiogenesis mechanism in skeletal muscle. This information has been added as new Supplementary Fig. 8 (page 18-19, line 409-426) and addressed in the revised Discussion (page 25, line 570-575).

21, “Given that *PGC1a* KO rescues the phenotype, why are all the differentially regulated cytokines identified in 5C tested in the *FNIP1*KO *PGC-1α*^{f/f}*MCK*Cre mice? A different list of genes are tested. This could be very informative to identify which factor secreted by the myocytes is needed to recruit immune cells.”

We have now added new data demonstrating that *Cxcl13* mRNA levels increase in FNIP1 KO muscles but are significantly suppressed in *Fnip1*^{-/-}, *PGC-1α*^{f/f}*MCK*-Cre muscles (revised Fig. 6d). All the differentially regulated cytokines identified in original Figure 5C are now tested in the *Fnip1*^{-/-}, *PGC-1α*^{f/f}*MCK*-Cre muscle. Many chemokine genes (e.g *Ccl5*, *Ccl8*, *Cxcl9*, *Cxcl10*, *Cxcl13*) are induced in FNIP1 KO muscles but suppressed in *Fnip1*^{-/-}, *PGC-1α*^{f/f}*MCK*-Cre muscles. One would predict that effective loss-of-function would be required targeting multiple of them.

Minor comments:

1. “The authors use the term “master regulator” on many occasions. That is a loaded term and should really be avoided. It implies that that factor can override all other factors, which rarely (or never) exists in biology.”

We have changed “master regulator” to “regulator” to be more accurate. Thanks.

2. “Spelling: agammaglobulinemic should be agammaglobulinemia.”

We have corrected this typographical error. Thanks.

3. “On line 136, the authors refer to “fig 1g,f”. It should be “1g,h”.”

It should be Fig. 1g,h instead of Fig. 1g,f. We apologize for this error, which has now been fixed.

4. “What are the units of figure 2i?”

The unit of Figure 2i is the Perfusion Unit (PU) for microcirculation blood perfusion. We thank the

reviewer and the units have been added to the Figure 2i.

5. *“The authors list the antibody for pericytes as “mPDGF Rb”. It should be mPDGFRb. But more importantly, the antibody is stated to be AF1402. That is a typo, it should be AF1042.”*

We have corrected this typographical error. Thanks.

REVIEWERS' COMMENTS

Reviewer #1 (Remarks to the Author):

In general the authors have performed the required additional studies. They also provided additional detail that was needed and/or requested.

Data on proliferation/apoptosis to link to the capillary density was requested and was added. Interestingly, it showed higher capillary density with higher apoptosis, with no difference in proliferation. This is the data but it is difficult to reconcile. By chance, have the authors looked at mice at older ages to see if the phenotype, the higher capillary density, goes away. In general skeletal muscle capillary density is a tightly regulated process and I wonder if the higher apoptosis is an attempt to correct the higher than expected/needed capillary density. Perhaps this could be added to the discussion.

Reviewer #2 (Remarks to the Author):

The authors have successfully addressed all my and other reviewer's comments. The methodologies are appropriate and statistical analysis are sound. The data supports their conclusion. This paper is worthy of publication in Nature Communications.

Reviewer #3 (Remarks to the Author):

The manuscript is much improved and I congratulate the authors for this.

I have one remaining concern. Throughout the figures, the number of capillaries assessed by CD31 staining is around 500 capillaries per mm² in control. The number of capillaries is with microsphere is around 40 capillaries per mm². But from the image, the vast majority of vessels have microspheres. I think there may be a calculation error here. If most vessels are perfused then the values between identification with CD31 and identification with microspheres should be almost the same. If this is not correct, could the authors explain the 10x difference in these numbers?

Point by Point Response to Reviewers' Comments**Itemized Responses to Reviewer #1**

“In general the authors have performed the required additional studies. They also provided additional detail that was needed and/or requested.

Data on proliferation/apoptosis to link to the capillary density was requested and was added. Interestingly, it showed higher capillary density with higher apoptosis, with no difference in proliferation. This is the data but it is difficult to reconcile. By chance, have the authors looked at mice at older ages to see if the phenotype, the higher capillary density, goes away. In general skeletal muscle capillary density is a tightly regulated process and I wonder if the higher apoptosis is an attempt to correct the higher than expected/needed capillary density. Perhaps this could be added to the discussion.”

We thank this reviewer for a comprehensive and insightful review. We would like to respectfully remind that our data actually showing lower capillary density with higher EC cell apoptosis in skeletal muscles from MCK-FNIP1 Tg mice (Fig. 1b,c and Supplementary Fig. 1d), and AAV9-based FNIP1 ablation in *Fnip1^{ff}* muscles leads to higher capillary density with lower EC cell apoptosis (Fig. 4j,k and Supplemental Fig. 5j,k). We do not believe that our data are difficult to reconcile. We agree with the Reviewer. It will be of interest to conduct future studies to investigate this FNIP1-dependent angiogenesis mechanism in skeletal muscle in the setting of aging. This point has been added to the revised Discussion (page 24, line 552-554).

Itemized Responses to Reviewer #2

“The authors have successfully addressed all my and other reviewer's comments. The methodologies are appropriate and statistical analysis are sound. The data supports their conclusion. This paper is worthy of publication in Nature Communications.”

We thank this reviewer for a comprehensive and insightful review.

Itemized Responses to Reviewer #3

“The manuscript is much improved and I congratulate the authors for this. I have one remaining concern. Throughout the figures, the number of capillaries assessed by CD31 staining is around 500 capillaries per mm² in control. The number of capillaries is with microsphere is around 40 capillaries per mm². But from the image, the vast majority of vessels have microspheres. I think there may be a calculation error here. If most vessels are perfused then the values between identification with CD31 and identification with microspheres should be almost the same. If this is not correct, could the authors explain the 10x difference in these numbers?”

We thank this reviewer for a comprehensive and insightful review. The Reviewer is correct. Our CD31 co-staining in microsphere injection experiment showed that the vast majority of vessels were perfused with microspheres, indicating that the blood vessels induced by FNIP1 deficiency are patent and capable of sustaining blood flow. Notably, as described in the Methods, longitudinal muscle cryosections were processed and subjected to CD31 co-staining for the visualization of perfused microvasculature, while transverse muscle sections were used for CD31 single staining. Our quantification of capillaries was normalized to mm², so the difference in the number of capillaries per mm² is a reflection of the different longitudinal vs. transverse quantification value per mm². We have added longitudinal and transverse labels to the revised Figures to clarify this point (revised Fig. 1g, Fig. 2d, Fig. 3e, Fig. 6h).